# Age-related differences in prefrontal glutamate are associated with increased working memory decay that gives the appearance of learning deficits

**Milena Rmus[1]\*, Mingjian He[2], Beth Baribault[1], Edward G Walsh[3], Elena K Festa[3], Anne GE Collins[1], Matthew R Nassar[3]**

[1]University of California, Berkeley, Berkeley, United States; [2]Massachusetts Institute of Technology, Boston, United States; [3]Brown University, Providence, United States

**Abstract** The ability to use past experience to effectively guide decision-making declines in older adulthood. Such declines have been theorized to emerge from either impairments of striatal reinforcement learning systems (RL) or impairments of recurrent networks in prefrontal and parietal cortex that support working memory (WM). Distinguishing between these hypotheses has been challenging because either RL or WM could be used to facilitate successful decision-making in typical laboratory tasks. Here we investigated the neurocomputational correlates of age-related decision-making deficits using an RL-WM task to disentangle these mechanisms, a computational model to quantify them, and magnetic resonance spectroscopy to link them to their molecular bases. Our results reveal that task performance is worse in older age, in a manner best explained by working memory deficits, as might be expected if cortical recurrent networks were unable to sustain persistent activity across multiple trials. Consistent with this, we show that older adults had lower levels of prefrontal glutamate, the excitatory neurotransmitter thought to support persistent activity, compared to younger adults. Individuals with the lowest prefrontal glutamate levels displayed the greatest impairments in working memory after controlling for other anatomical and metabolic factors. Together, our results suggest that lower levels of prefrontal glutamate may contribute to failures of working memory systems and impaired decision-making in older adulthood.

\*For correspondence:
milena_rmus@berkeley.edu

**Competing interest:** The authors declare that no competing interests exist.

## Editor's evaluation

This important study combines behavior, computational modelling and magnetic resonance spectroscopy in a cross-sectional design to address the question of whether age-related differences in learning are driven by changes in working memory decay or deficiencies in the reinforcement learning (RL) system. The general approach is convincing, the data novel, and the analysis carefully executed. Future work requires a longitudinal design to separate aging from cohort effects and may address the generality of the effects to other RL/Working memory tasks.

## Introduction

People and animals undergo a number of cognitive changes across healthy aging, and while some of these changes reflect improvements emerging from an extended lifespan of continual learning, others are characterized by declines in function with advanced age (*Cattell, 1943*). In particular, recent work has highlighted age-related deficits in decision-making under uncertainty, particularly in situations where decision outcomes are learned through experience (*Eppinger et al., 2013*). While

such deficits could have critical importance for both aging individuals and society, the exact mechanisms underlying these deficits remain unclear. One possibility is that such impairments stem from an inability to retain the task-relevant information (i.e. working memory, *Salthouse and Babcock, 1991*), whereas another possibility is that such impairments stem from failures to learn state action values through reinforcement (i.e. reinforcement learning, *Chowdhury et al., 2013*). These different cognitive processes are supported by different neural systems. Understanding how they contribute to impairments in decision-making across lifespan could also help shed light on biological age-related mechanisms of impaired decision-making.

Previous work has shown that working memory declines with aging (*Salthouse and Babcock, 1991*). Such declines have been shown in both cross-sectional and longitudinal studies and across a range of working memory tasks (*Salthouse and Babcock, 1991*). Working memory is thought to be implemented through persistent activation of neural firing in similarly tuned neurons within recurrent excitatory networks in prefrontal and parietal cortices (*Goldman-Rakic, 1995*; *Andersen and Buneo, 2002*). Within such networks, successful maintenance of activity over a delay period depends on glutamatergic signaling, and in particular activation of NMDA receptors (*Durstewitz et al., 2000*; *van Vugt et al., 2020*). In aged monkeys, such persistent activations in prefrontal cortex are diminished and can be restored through local pharmacological manipulations that affect intrinsic neural excitability (*Wang et al., 2011*). While the mechanisms by which persistent activity deteriorates with age are not fully established, observed reductions in synapses and dendritic spines on aged prefrontal pyramidal neurons (*Dumitriu et al., 2010*) suggest that they result at least in part from an overall reduction in glutamatergic signaling. Thus, if age-related behavioral deficits stem from diminished working memory functions, they may result from reductions in glutamate signaling in prefrontal and parietal regions that support working memory.

Reinforcement learning (RL) systems are also thought to decline across healthy aging. Evidence for such declines comes from behavioral impairments in tasks where participants learn to map stimuli onto rewarded actions (*Eppinger et al., 2013*; *Frank and Kong, 2008*; *Hämmerer et al., 2011*). Learning in such tasks is thought to occur through reinforcement of associations between stimulus and action in the striatum driven by dopamine reward prediction error signals (*Frank et al., 2004*; *Collins and Frank, 2014*). Such reward prediction error signals are blunted in healthy aging (*Samanez-Larkin et al., 2014*) supporting the accounts of age-related decline in RL (*Chowdhury et al., 2013*; *Eppinger et al., 2013*). *Chowdhury et al., 2013* found that boosting dopamine signaling with dopamine precursor levodopa (L-DOPA) led to restoration of RPEs and recovery of learning performance in older adults. The results from these studies appear to indicate that neural mechanisms underlying reinforcement learning are impaired in older adults; however, a number of other studies have suggested that RL systems are intact in older adults (*Radulescu et al., 2016*; *Grogan et al., 2019*). Some of the discrepancy in these results may stem from inconsistencies in the tasks used to measure RL across different studies (*Eckstein et al., 2022*) as many tasks that could be solved through incremental learning in the striatum according to reward prediction errors (RL) could also be solved using other cognitive systems, working memory systems being one of particular importance (*Yoo and Collins, 2022*).

Typical reinforcement learning tasks require identifying the best action to choose when confronted with a given stimulus based on previous feedback. While RL models of such tasks assume that action values are learned incrementally, as if through adjustment of synaptic weights in the striatum, it is also possible for participants to achieve success on such tasks through short-term storage of recent trial information in working memory. Recent work has used task designs that can dissociate these potential contributors to goal directed behavior, along with models that can quantify them, to show that behavioral deficits previously thought to reflect reinforcement learning were actually attributable to working memory systems (*Collins et al., 2014*). The reinforcement learning–working memory task (*Collins and Frank, 2012*; *Collins, 2018*; *Collins et al., 2014*; *Collins and Frank, 2018*) is a simple stimulus–response association learning task with a format that is commonly observed in RL studies (*Frank and Kong, 2008*). However, this task includes a WM manipulation – varying the number of stimulus–response actions (set size) participants need to learn. This manipulation targets WM as a capacity-limited, short-term process since increasing set-size increases both load and average duration between stimulus repetitions. Thus, varying the set size can shift contribution of RL/WM to performance and help tease apart which of the two participants are relying on when performing the task. This task has yielded important mechanistic

insights into general as well as clinical populations (*Collins et al., 2014*; *Collins and Frank, 2012*; *Master et al., 2020*).

In this study, we combined cognitive, computational, and neural approaches to address the question of (1) how the age-related deficits in making decisions from experience emerge from RL and WM computational mechanisms and (2) how the differences in these mechanisms relate to age-related changes in relevant neural systems. To this end, we administered an RL-WM task to young and older adults and modeled their behavior using computational model that can distinguish between deficits in RL and WM systems. We used magnetic resonance spectroscopy (MRS) to measure levels of glutamate and GABA in regions thought to support working memory (prefrontal/parietal cortices) and reinforcement learning (striatum).

We found that older participants performed worse in the RL-WM task than younger adults. Group comparison of model parameters revealed that older participants had a more rapid decay of task relevant information in working memory. Older adults also had reduced glutamate levels, particularly in prefrontal and parietal cortices where glutamate is thought to support the persistent activation that underlies working memory. Furthermore, reductions in prefrontal glutamate were correlated with working memory decay across individuals – such that those with the lowest levels of prefrontal glutamate had the most rapidly decaying working memories. Taken together, our results suggest that age-related working memory differences likely contribute to decision-making deficits in older adults, and that such deficits may result from failures of recurrent excitatory networks to sustain persistent representations due to reduced levels of glutamate.

## Results

42 older (age mean [SD] = 68 [8.5]) and 36 younger (age mean (SD) = 21 [4.4]) participants were enrolled in a two-session study. Age-normed total *The Repeatable Battery for the Assessment of Neuropsychological Status* (RBANS) (*Randolph et al., 1998*) scores were similar in both groups (mean [std] younger: 106.4 [12.0], older: 109.0 [10.8]), suggesting that our cohorts reflected comparable samples of the population with respect to overall cognitive ability. Furthermore, all participants scored in cognitively healthy range (younger adults range: 83–140; older adults range: 81–141) since cognitive impairment is defined as score 70 or below. The first session required performance of an RL-WM task designed to dissociate the contributions of working memory and reinforcement learning to decision-making under uncertainty. The second study session included an MRI session in which MRS was used to measure glutamate and GABA in key regions thought to support working memory (middle frontal gyrus [MFG] of prefrontal cortex and intraparietal sulcus [IPS]) and reinforcement learning (striatum).

In the RL-WM task, older and young adults were required to learn correct stimulus–response (key press) associations, with the goal of earning as many points as possible. After the stimulus appeared, the participants had 1 s to make their response, following which they received deterministic feedback. Each stimulus appeared nine times in a block, pseudo-randomly interleaved with other stimuli, allowing the participants to learn correct associations from feedback. The task has a common RL experiment structure (*Frank et al., 2004*), with an important difference: the number of associations varied (either three or six) between different, fully independent blocks. This enabled us to manipulate the degree to which working memory (WM), a capacity-limited system, might contribute to behavior by storing recent stimulus–action–outcome information. In particular, previous computational and empirical work using this task suggests that working memory will contribute less to decisions in the high set size condition (six associations), but that slower learning in high set size paradoxically leads to better retention relative to learning in a surprise 'test phase' in which learned associations are tested in the absence of feedback (*Collins, 2018*).

### Behavioral results; model-independent

#### Learning

Both age groups learned to make accurate responses in the learning phase and did so more quickly in blocks with fewer interleaved stimuli, but younger adults learned more successfully than their older counterparts (*Figure 1B and C*). The learning curves reveal patterns that are consistent with RL and WM involvement in young adults (early spike in accuracy for lower set size that is absent for larger set size and that is considered a marker of fast/one-shot learning characteristic of WM but not RL; *Collins*

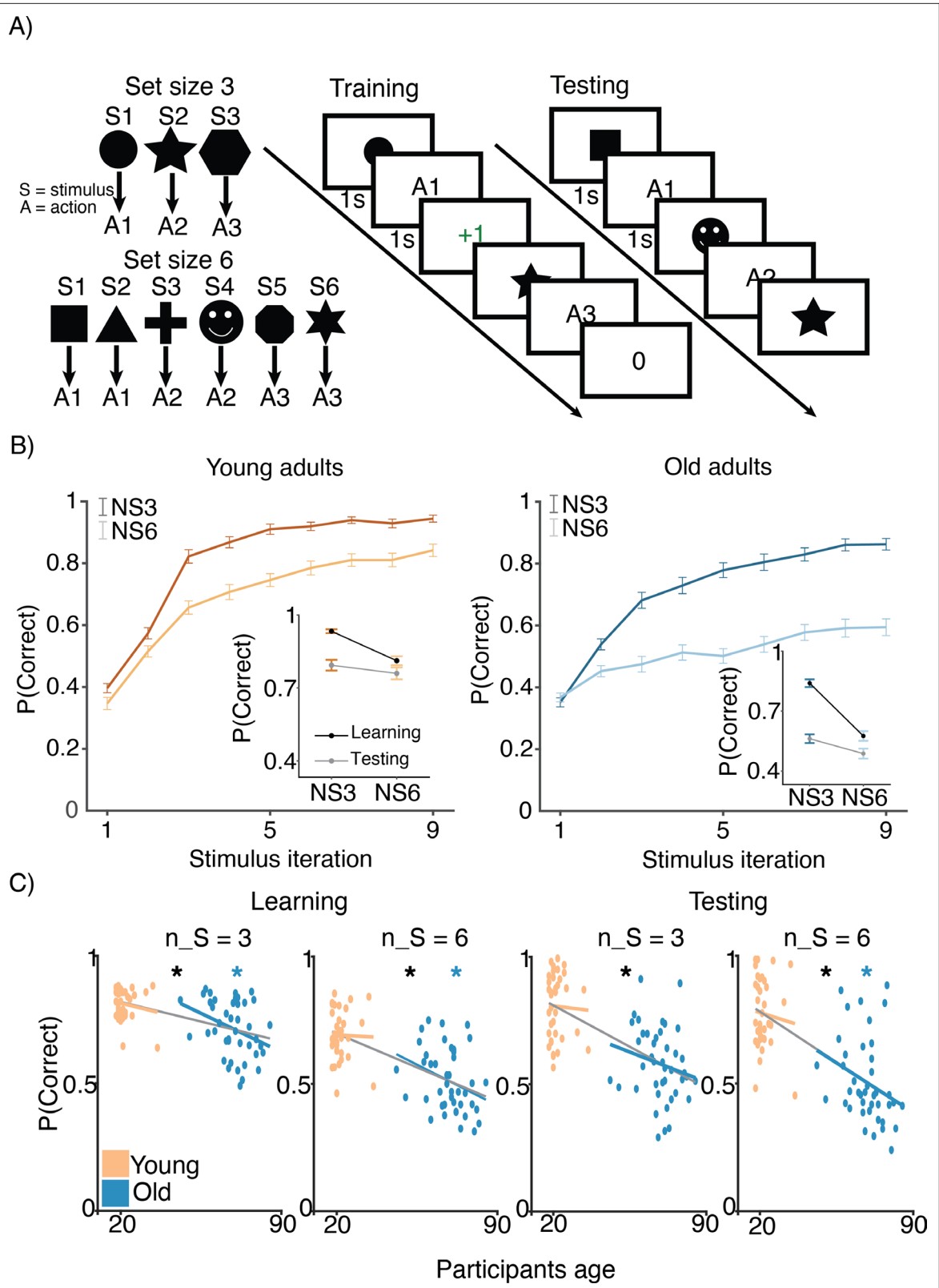

**Figure 1.** Experimental design. (**A**) Reinforcement learning–working memory (RL-WM) task with $n_S = 3$ and $n_S = 6$ blocks. Participants learned three or six stimulus–action associations and received truthful feedback on each trial. In the test phase, participants observed the same images observed during learning and were asked to produce the responses they remembered being correct, without being given the feedback. (**B**) Learning curves and learning/testing comparison for both set sizes in younger (N=36) and older adults (N=42). Younger adults performed better overall, with smaller difference in

*Figure 1 continued on next page*

*Figure 1 continued*

performance between the set sizes. The older participants showed greater drop-off in set size 3 between learning and testing compared to young adults. Error bars represent standard error of the mean (SEM). (**C**) Age spearman correlations with average accuracy in each set size conditions across both learning and testing. All full sample negative correlations between age and performance were significant (gray lines); there were no significant age–performance correlations within young age group (yellow lines). Age and performance were significantly correlated within old group (blue) in all conditions except set size 3 in testing. The correlation significance was evaluated using the p < 0.05 threshold.

*and Frank, 2012*). This pattern is less clear in older adults. The average accuracy of individuals in the young and old groups was 0.74 (SD = 0.01) and 0.60 (SD = 0.01), respectively (ANOVA for group difference: $F(1, 76) = 45.42, p = 2.71e − 09$). Younger adults had higher accuracy compared to older adults for both set sizes 3 ($t(76) = 4.71, p = 1.05e − 05$) and 6 ($t(76) = 7.03, p = 7.61e − 10$). Furthermore, we confirmed that the difference between set size 3 and set size 6 performance was greater in older adults ($t(76) = 4.12, p = 9.55e − 05$). Training performance deficits in both set sizes were greatest in the oldest individuals, as revealed by significant correlations between age and training accuracy within the older group (*Figure 1C*; $r(n_S = 3\ accuracy, age) = −0.42$, $p = 0.03$; $r(n_S = 6\ accuracy, age) = −0.41$, $p = 0.04$).

To test what factors (i.e. set size, reward history, delay) impact accuracy on a trial-by-trial level, we ran a mixed-effects general linear model (GLM). The GLM analysis of accuracy data indicated that both groups showed signatures of RL and WM in their behavior. We found a positive effect of reward history ($\beta = 1.74$, $t = 43.51$, $p < 0.0003$) that was apparent in both young and old groups (young: $\beta = 1.66$, $t = 27.60$, $p < 0.0003$ ; older: $\beta = 1.74$, $t = 32.5$ , $p < 0.0003$). We also found negative effects of set size and delay on accuracy (set size: $\beta = −0.26$, $t = −9.91$, $p < 0.0003$ ; delay: $\beta = −0.30$, $t = −13.3$, $p < 0.0003$) that were apparent in both young (set size: $\beta = −0.23$, $t = −7.55$, $p < 0.0003$; delay: $\beta = −0.21$, $t = −5.11$, $p < 0.0003$) and older adults (set size: $\beta = −0.34$, $t = −9.73$, $p < 0.0003$; delay: $\beta = −0.29$, $t = −8.92$, $p < 0.0003$).

Coefficients from the GLM, particularly those capturing behavioral hallmarks of working memory (i.e. set size, delay), could be used to infer participants age. A linear regression to predict the individual participants' age based on their respective coefficients from the behavioral logistic regression model yielded negative coefficients for the accuracy fixed effect ($\beta = −17.2$, $t(71) = −5.64$, $p = 3.1455e − 07$) and its sensitivity to set size ($\beta = −7.44$, $t(71) = −3.01$, $p = 0.0035$), suggesting that older adults had lower overall performance but were also more affected by the set size manipulation. Reward history coefficients took positive values ($\beta = 12.73$, $t(71) = 4.24$, $p = 6.5262e − 05$) in the same model, suggesting that older adults were also more sensitive to previous rewards than their younger counterparts.

## Testing

Both age groups experienced set size-dependent declines in performance during the test phase of the task, with older adults achieving lower levels of performance overall (*Figure 1B*, inset figures). Consistent with the notion that greater use of WM during learning in small set sizes weakens RL-dependent learning as expressed in the test phase (*Collins, 2018*), participants had greater declines in performance for set size 3 than for set size 6 between training and testing (t($\Delta_{3^L,3^T}$, $\Delta_{6^L,6^T}$) young: $t(35) = 4.64$, p < 0.001 ; older: $t(41) = 8.1$, p < 0.001). The set size-based asymmetry in accuracy decline from training to testing was numerically greater in older adults, but the effect was not significant ($t(76) = 1.62$, p = 0.1).

A GLM fit to test phase accuracy confirmed that reward history was still strongly linked to performance in the test phase, but that set size was less important. An analogous mixed effect model to that used on the training data revealed positive reward history effects in both young and older adults (young: $\beta = 1.03$, $t = 16.2$, p < 0.001; older: $\beta = 0.92$, $t = 0.92$, p < 0.001). In contrast, in the test phase there was limited evidence for an effect of set size on performance in young adults ($\beta = −0.15$, $t = −1.85$, $p = 0.06$) and only a very modest effect in the older adults ($\beta = −0.20$, $t = −3.07$, p = 0.002).

In summary of our basic behavioral results, general patterns in the data are consistent with contributions of both working memory and reinforcement learning to behavior, replicate previous findings showing interactions between the two revealed in the test phase, and confirm age-related variability in performance. To better understand how the WM and RL mechanisms vary with age, we

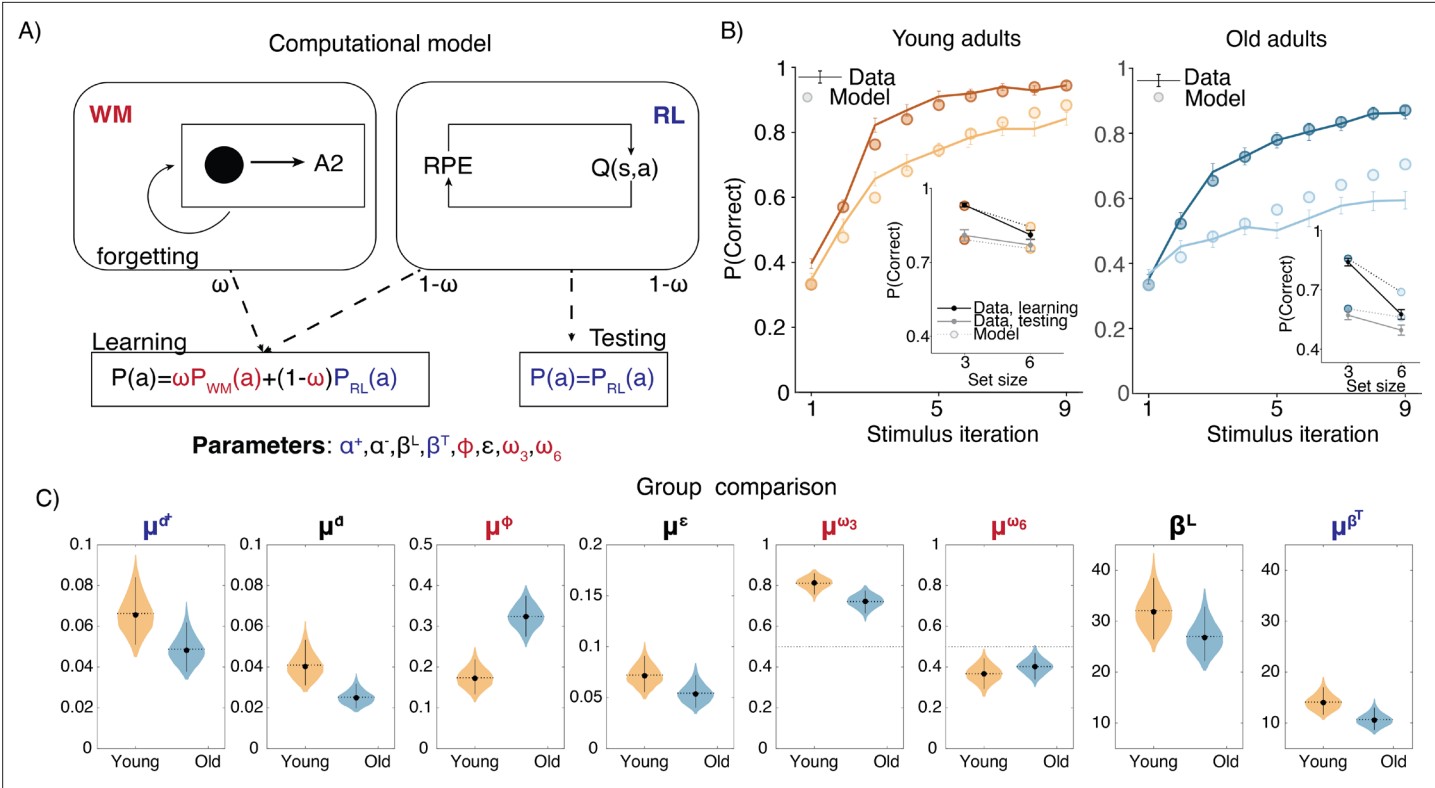

**Figure 2.** Computational model. (**A**) Reinforcement learning–working memory (RL-WM) computational model schematic. (**B**) Posterior predictive checks for group-level learning curves and asymptotic (last three stimulus iterations) means from our Bayesian RL-WM model. Darker color represents set size 3; lighter color represents set size 6. (**C**) Comparison of group-level mean parameters shows that primarily WM parameters drive differences between young and older adults, especially the forgetting rate parameter $\mu^{\phi}$. Dashed lines represent equal contributions of WM and RL processes.

used computational modeling to further dissect the computational factors giving rise to differences in participant behavior.

## Modeling results

To gain more insight into how specific learning mechanisms vary with age, we fit a computational model to the RL-WM task behavioral data. Specifically, we applied a hybrid RL-WM model that quantifies how WM and RL contribute to learning together (***Figure 2***; for a detailed description of how the model captures WM and RL processes see 'Methods' section). Parameters attributed to WM consisted of (1) WM decay ($\phi$) – capturing the rate at which stimulus–response associations decayed during learning, and (2) set size-dependent WM reliance parameters ($\omega^3$ and $\omega^6$) – capturing relative reliance on WM (as opposed to RL) to make choices. RL parameters consisted of the learning rate ($\alpha^+$) and test phase softmax inverse temperature ($\beta^T$), governing the rate at which participants updated the stimulus–response association values based on observed feedback, and the extent to which participants' test phase choices were deterministic vs. exploratory, respectively. Learning phase softmax beta ($\beta^L$) captured the same dynamic during the learning phase (but unlike all other RL-WM model parameters was estimated at the group-level only; see 'Methods' for details). We also incorporated a negative learning rate ($\alpha^-$) to differentiate learning from negative versus positive feedback, which was present in both RL and WM modules of the model. Finally, the model included a noise parameter in response selection ($\epsilon$) to capture random lapses in task performance.

We implemented the RL-WM model in a hierarchical Bayesian framework, as Bayesian model fitting offers many statistical benefits (e.g. ***Kruschke and Liddell, 2018***) and incorporating hierarchical structure allowed us to account for individual- and group-level effects simultaneously (among other practical benefits; e.g. ***Lee, 2011***). As this represents a novel extension of the RL-WM model, we offer full model specification in the 'Methods' section; further details on the exact approach to Bayesian model fitting are also available in 'Methods.

## Model comparison

We first compared the performance among the three versions of our Bayesian RL-WM model (with different hierarchical structures; see 'Methods' for details) in order to select a model to use in all model-based analyses of our data. Our model comparison indicated that the 'two-group' version of the model, which incorporated separate hierarchies over the participants within each age group, offered the best description of our data (see *Appendix 1—figure 2*). This two-group, hierarchical version of the model outperformed both the non-hierarchical model ($\Delta \mathrm{WAIC} = 1.62 \times 10^3$) and the model that included a single hierarchy over all participants, regardless of their age group ($\Delta \mathrm{WAIC} = 1.86 \times 10^2$). This model's posterior predictive learning curves and posterior predictive asymptotic means (*Figure 2B*) successfully captured the general patterns in performance of both age groups on the RL-WM task.

## Model parameter recovery

We used model parameter recovery to ensure that parameters in our best fitting model are separable, and that our model is sensitive to individual variability in model parameters. We ran a recovery study where we simulated data from estimated parameters according to the RL-WM model using the same task design as in our human experiment and then fit the synthetic data with our model. We confirmed that both individual-level parameters and group-level hyperparameters were recovered successfully (see *Appendix 1—figure 9*).

## Parameter analysis

We next examined which parameters of the model differed substantially across age groups. The strongest difference between the age groups was with respect to decay rate $\phi$ in the WM module. The group mean decay rate was much higher in the older age group ($\Delta \mu^\phi = -0.151$, 95% equal-tailed credible interval [CrI] = [-0.227, -0.070]) indicating that the stimulus–action–outcome associations stored in WM degraded faster for older adults. The mean decay rate for the younger group was 0.174, indicating that an association would decay to 83% of its original strength after a single trial, whereas in older adults the decay of 0.324 would lead to working memory degradation about twice as fast.

Next, we focused on the relative WM reliance parameter $\omega$. We were interested in exploring whether the WM reliance might follow the similar pattern observed in group-related difference in the WM decay. For instance, if older participants' WM module is markedly more forgetful, they would accordingly rely less on the association weights stored in WM to guide their action selection. In our analysis, we observed that the relative reliance on WM over RL — as captured by the set size-dependent policy mixture parameters $\omega^{n_S}$ — depended heavily on the set size in both groups. While learning stimulus–action associations in set size 3, participants in both groups relied more on WM ($\omega^3 > 0.5$ for 36 of 36 younger and 40 of 42 older participants). In set size 6, participants in both groups tended to rely more on RL ($\omega^6 < 0.5$ for 32 of 36 younger and 35 of 42 older participants). Importantly, $\omega^3$ was greater than $\omega^6$ for 36 of 36 younger and 40 of 42 older participants. We observed the same effect for the group mean parameters, with similar magnitudes for both groups ($\mu^{\omega^3} - \mu^{\omega^6}$, young: [0.336, 0549]; older: [0.214, 0.420]). Older adults had a somewhat reduced tendency to rely on WM ($\Delta \mu^{\omega^3} = 0.090[-0.003, 0.179]$) compared to younger adults, even when doing so is appropriate (i.e. for the smaller set size), but this difference was not robust. Thus, older and young adults showed similar patterns in WM reliance, contingent on the set size.

The model fits revealed very little indication of group differences in parameters specific to the RL system. We found no group differences in the test phase beta $\beta^T$ ([-0.465, 7.673]) or the learning rates $\alpha$ ([-0.007, 0.042]). Our model did have a learning rate asymmetry parameter that affected both RL and WM systems, and allowed the model to capture a consistent trend toward higher learning from positive relative to negative outcomes in both groups. This was evident in group means ($\mu^{\alpha^+} - \mu^{\alpha^-}$, young: [0.004, 0.049], older: [0.009, 0.0041]) as well as for individual participants ($\alpha^+ > \alpha^-$ for 34 of 36 young and 42 of 42 older participants). This asymmetry differed across groups, such that negative learning rates were higher in the young adult group compared to the older adults (*Figure 2C*) ($\Delta \mu^{\alpha^-}$ = 0.016, [0.002, 0.032]). This suggests that older adults neglected negative feedback even more relative to young adults. This result is inconsistent with the previous work, suggesting that older adults exhibit bias in learning from negative outcomes (*Frank and Kong, 2008*). However, this finding has not been consistent across studies, and in our paradigm a bias toward learning from positive outcomes could

be viewed as an adaptive strategy when resources are limited, in that positive outcomes perfectly prescribe the correct future action, whereas negative ones only rule a single response out.

Importantly, the noise/random lapse parameter did not differ between the two age groups ($\Delta \mu^\epsilon = [-0.011, 0.047]$), which suggests it is unlikely that the observed differences in accuracy are due to older adults simply having noisier data.

To demonstrate the impact of their higher decay rate on older adults' learning, we again simulated older adults' behavior from the fitted model, except we used the young adults' group-level decay rate in place of every older adult's individual-level decay rate. To do this, we used the same procedure as was used to generate the posterior predictive plots (as in *Figure 2b*; see 'Methods'), except with the aforementioned parameter swaps. This demonstration (see *Appendix 1—figure 5*) suggests that the older adult's higher decay rate could potentially account for most of the difference in learning behavior between the age groups.

Our behavioral results revealed that older adults performed more poorly in set size 6, in which performance is more driven by RL than in set size 3, due to capacity limitations of WM. However, our modeling revealed that WM parameters, specifically WM decay $\phi$, show greatest group differences, rather than RL parameters. Simulation results (see *Appendix 1—figure 5*) provided some insight into this finding by showing that decreasing the working memory decay in older adults improves training performance most notably in the set size 6 condition. The primary explanation for this is that higher set size assumes increased delay between successive presentations of the same stimuli. Longer delays, in turn, make differences in WM decay more pronounced. Furthermore, a WM impairment can have an indirect effect in RL, in that frequent failure to select correct action through WM leads to reduced ability to train RL on encoding correct responses (especially earlier in training, when the incremental RL has not 'caught up' yet), and thus worse performance overall. Thus, our behavioral and modeling results are not incongruous and simply highlight the importance of modeling as it allows us to identify the mechanisms that drive behavioral patterns with greater specificity (*Radulescu et al., 2019*; *Collins and Frank, 2012*).

## Linking MRS measures, model parameters, and behavioral performance

Thus far, our results suggest that (1) there is a significant difference in performance between two age groups, and (2) that this difference is best explained by working memory deficit in older group (most significantly WM decay, and marginally reliance on WM in smaller set size condition). Next, we aimed to test how these age-related differences relate to metabolic changes affecting brain neurochemistry. We used MRS to measure neurotransmitters (glutamate and GABA) that support working memory (*Wong and Wang, 2006*) in regions important for both reinforcement learning (striatum [STR]) and working memory (middle frontal gyrus of prefrontal cortex [MFG], intraparietal sulcus [IPS]). To control for larger structural changes that may covary with differences in glutamate and GABA levels, we included gray volume, white matter volume and cortical thickness (in caudal middle frontal [CMF], superior frontal [SF], and rostral middle frontal [RMF]) cortex in our analyses and to control for nonspecific metabolic changes we included measures of N-acetylaspartate (NAA).

In order to understand how differences in performance and their computational mechanisms relate to the neural measures. we took a data-driven multistep analysis approach. First, we constructed a GLM to explain overall learning performance according to structural measures collected as part of our high-resolution anatomical MRI scans (white matter, gray matter, cortical thickness), as well as neurochemical measures from MRS (glutamate, GABA, NAA). Given the correlations in our neurochemical measures across brain regions, we attempted to maximize power to detect relationships in our initial linear model by aggregating neurochemical measures across brain regions. Specifically, we averaged the MRS measures across regions (i.e. average glutamate was the average of glutamate measures from medial frontal gyrus, intraparietal sulcus, and striatum). Anatomical specificity of significant predictors was then examined in follow-up analyses. We set up a linear model predicting average learning performance using the following z-scored predictors: average GABA, glutamate, NAA, gray matter, white matter, and cortical thickness in caudal middle frontal (CMF), superior frontal (SF), and rostral middle frontal (RMF) areas (cortical thickness measure was not averaged across the three areas because these measures were selected a priori from a Freesurfer parcellation and are unrelated to the voxels; see 'Methods' for details). The best model identified through a regularized and cross-validated fitting procedure (see 'Methods') included the following predictors: GABA, glutamate, NAA, CMF

cortical thickness, and SF cortical thickness ($R^2 adjusted = 0.41$, $F = 7.8$, $p < 0.001$, variance inflation factors: [1.16, 2.06, 1.76, 2.71, 2.37]). However, out of all the predictors in this model, the only coefficient within this model for which a value of zero could be reliably rejected was our aggregate measure of glutamate concentration ($\beta = 0.04$, $t = 2.5$, p = 0.01; **Figure 3B**).

Next, we used our best-fitting model to generate brain-based predictions of performance so that we could evaluate their computational specificity. Note that these predictions depended on all biological factors in the model with best out-of-sample accuracy, which outperformed simpler models, including one that only had glutamate as a predictor. These predictions can be thought of as projections of behavioral learning performance onto the axis of brain measures most closely related to it. We then fit a linear model that regressed the brain-predicted performance onto an explanatory matrix that contained all parameter estimates from our behavioral model in order to determine which computational elements are most tightly linked to the neural fingerprint of performance (or performance failures). The model had the following structure: brain-predicted performance ~1 + all RL-WM parameters. This model explained a significant amount variance in our brain-based performance measure ($R^2_{\text{adjusted}} = 0.48$, $F = 7.46$, p < 0.001) and revealed that WM decay $\phi$ ($\beta = 0.02$, $t = -2.24$, p = 0.03) and set size 3 WM weight $\omega_3$ ($\beta = 0.02$, $t = 2.35$, p = 0.03) contributed substantially to these predictions, suggesting that the brain-based predictions largely reflected the integrity of a working memory system (**Figure 3C**).

Having established that (1) glutamate contributes to accuracy and that (2) WM parameters $\phi$ and $\omega_3$ capture the performance predictions based on neural measures (including glutamate), we next tested the anatomical specificity of the glutamate–working memory relationship (see 'Methods'). To do so, we constructed two additional regression models, in which we regressed $\phi$ ($R^2 adjusted = 0.32$, $F = 9.22$, p < 0.001) and $\omega_3$ ($R^2 adjusted = 0.23$, $F = 6.42$, p < 0.001) onto three separate glutamate measures extracted from MFG, IPS, and striatum (model 1: $\phi$ 1 + MFG glutamate + IPS glutamate + STR glutamate; model 2: $\omega_3$ 1 + MFG glutamate + IPS glutamate + STR glutamate). From these two models, the only significant coefficient was the effect of MFG glutamate on WM decay $\phi$ ($\phi$: MFG glutamate $\beta = -0.04$, $t = -2.95$, p = 0.004; STR glutamate $\beta = -0.008$, $t = -0.66$, p = 0.50; IPS glutamate $\beta = -0.023$, $t = -1.67$, p = 0.10; $\omega_3$ : MFG glutamate $\beta = 0.02$, $t = 1.48$, p = 0.14; STR glutamate $\beta = 0.02$, $t = 1.77$, p = 0.08; IPS glutamate $\beta = 0.02$, $t = 1.70$, p = 0.09) (**Figure 3D**). Specifically, lower glutamate levels were associated with higher WM decay. This relationship was specific to glutamate and was not observed when we substituted measures of glutamine ($r = -0.05$, p = 0.70), a molecule with a nearby spectral peak. The relationship between MFG glutamate and WM memory decay persisted even after regressing out variance related to MFG gray matter (GM) and MFG creatine (Cr) ($r = -0.39$, p = 0.0035), suggesting that our results are specifically related to glutamate rather than picking up on nonspecific anatomical differences such as gray matter or tissue density. For completeness, we also tested the correlation between MFG glutamate and other model parameters. We found that MFG glutamate also correlated with $\alpha$, $\omega_3$ and $\alpha^-$ (see **Appendix 1—figure 7**). However, when these parameters were included in a model that also contained the WM decay parameter (MFG glutamate ~ 1 + $\phi$ + $\alpha$ + $\omega_3$ + $\alpha^-$), only $\phi$ was a significant predictor ($\phi$ : $\beta = 0.24$, $t = -3.14$, p = 0.002, $\alpha$ : $\beta = -0.03$, $t = -0.42$, p = 0.67, $\omega_3$ : $\beta = 0.12$, $t = 1.84$, p = 0.07, $\alpha^-$ $\beta = 0.04$, $t = 0.56$, p = 0.57). Taken together, these results suggest that higher levels of WM decay were associated with lower prefrontal glutamate levels.

Next, we examined how age factors into this relationship. We found that age negatively correlated with glutamate ($r_{\text{Spearman}} = -0.59$, $p = 2.93e - 06$) and positively correlated with decay ($r_{\text{Spearman}} = 0.63$, $p = 3.15e - 07$). The visualization of this depicted in **Figure 3D** suggests that older adults have lower glutamate levels and higher decay, whereas inverse is true for younger adults. Since age was negatively correlated with glutamate and positively correlated with decay, there is a possibility that age alone could fully explain the correlation between glutamate and WM decay. To address this question, we (1) ran a model predicting WM decay using MFG glutamate and age, as well as age-by-MFG glutamate interaction, and (2) correlation between MFG glutamate and WM decay in the two age groups separately. The interaction was not significant ($\beta = -0.01$, $t = 0.69, p = 0.48$), and the main effects of glutamate and age were not significant when entered together in the model (age: $\beta = 0.14$, $t = 1.24, p = 0.21$; glutamate: $\beta = -0.02$, $t = -1.03, p = 0.30$), likely due to the fact that glutamate and age are correlated. We found that glutamate did not correlate with WM decay in young adults ($r_{\text{Spearman}} = 0.11$, p = 0.54), but there was a trending correlation between glutamate and WM decay

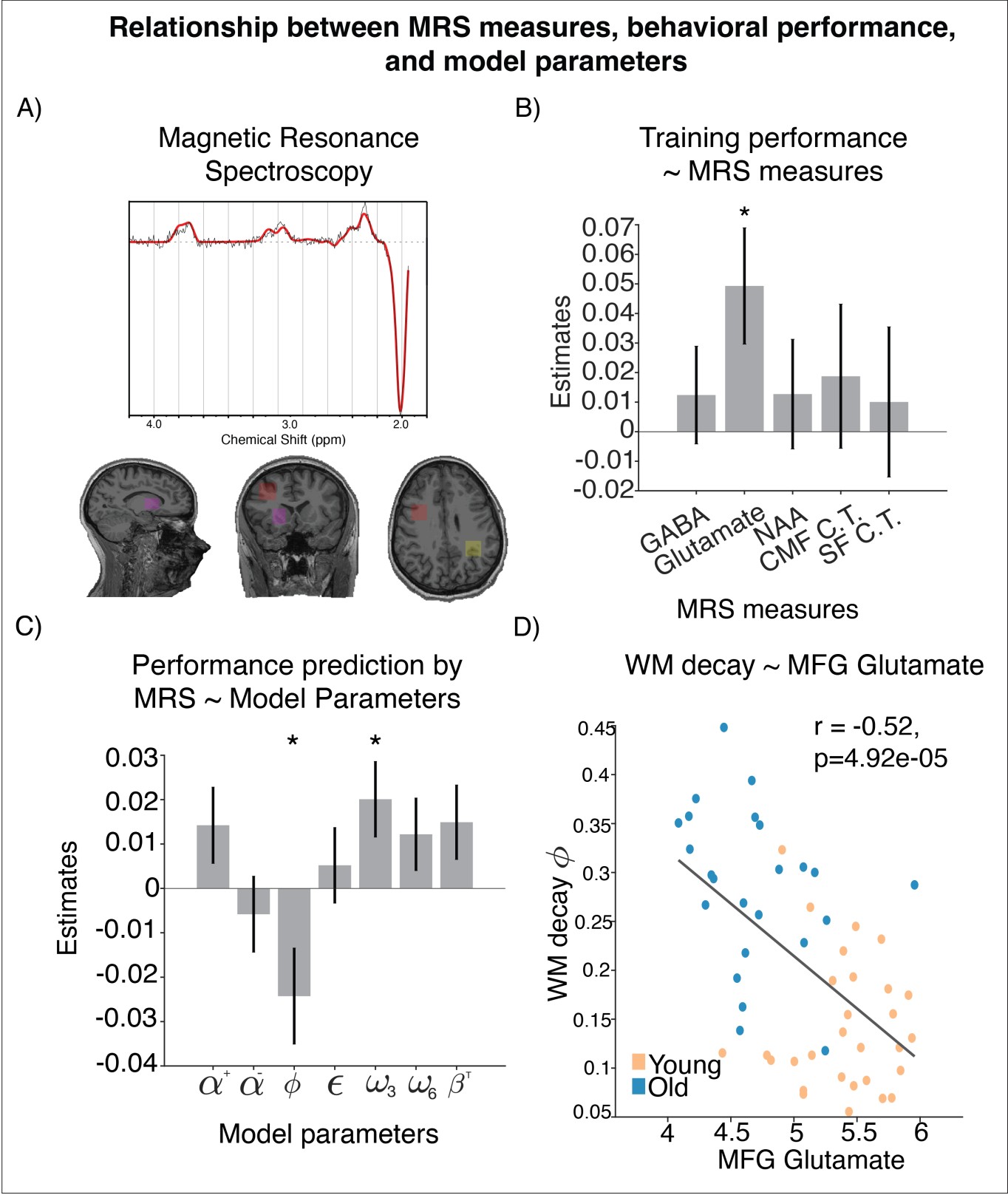

**Figure 3.** Relationship between magnetic resonance spectroscopy (MRS) measures, behavioral performance, and model parameters. (**A**) Location of voxels from which the spectroscopy measures were sampled. (**B**) Neural measure predictors that provided best out-of-sample prediction of learning performance. (**C**) Coefficients of modeling brain-predicted performance using all reinforcement learning–working memory (RL-WM) computational model parameters in a linear model (N=53). WM decay and WM weight in set size 3 are the only significant predictors. Errorbars represent standard

*Figure 3 continued on next page*

*Figure 3 continued*

error of the coefficients. (**D**) Relationship between WM decay and glutamate levels in middle frontal gyrus (MFG), with Spearman correlation shown for reference.

in older adults ($r_{Spearman} = -0.36, p = 0.08$). Finally, we used a formal mediation model to examine whether the relationship between age and working memory decay was weakened after accounting for glutamate. While the sign of the mediating effect in the model was consistent with glutamate reducing the explanatory effects of age, the effect was not large enough to reject the null hypothesis that glutamate does not mediate the effects of age on working memory ($ACME = 0.009, p = 0.28$, proportion mediated = 0.13). Taken together, these results reiterate that age is closely associated with working memory decay and reduced glutamate levels, but provide very little conclusive evidence regarding the exact nature of these relationships.

## Discussion

Our results suggest that age-related differences in the ability to make decisions from experience are driven by increased forgetting in working memory that is linked to underlying levels of glutamate in prefrontal cortex. Our results provide a computationally specific account of learning impairments that have previously been observed over the course of healthy aging, and also provide insight into its biological underpinnings, in particular suggesting that reduced glutamate impairs the ability of prefrontal recurrent networks to maintain task relevant information over extended periods of time.

While it has long been established that working memory declines over healthy aging (*Salthouse and Babcock, 1991*), our work shows that these deficits are also at the root of age-related impairments in decisions from experience (*Eppinger et al., 2013*). Importantly, this result held even when simultaneously accounting for RL contributions to behavior, which have also been theorized to be impacted by cognitive aging (*Frank and Kong, 2008*; *Li et al., 2015*). As shown by examples of previous work (*Collins et al., 2014*; *Leong et al., 2017*; *Radulescu et al., 2019*), it is important to consider the interactions between neurocognitive systems rather than studying them in isolation as this could lead to an incorrect attribution of observed behavioral patterns. Our behavioral modeling results are complementary to those of another recent study that leveraged varying delay between training and testing to examine contributions of working memory, and concluded that older adults engaged WM less than young adults during the short delays (*van de Vijver and Ligneul, 2020*). Our results indicated that older adults had more rapid decay of task relevant working memories, but did not substantially reduce their deployment for selecting actions. Thus, across studies, there is converging evidence that working memory deficits in older adults can negatively impact decisions from experience, perhaps providing insight into why decisions from description are relatively spared by the cognitive aging process as such decisions do not typically require working memory (*Li et al., 2015*). However, our results imply that older adults did not compensate for their unreliable memory system by using working memory less, thereby increasing the impact of working memory deficits on behavior. Though our framework provided an ideal setup to disentangle contributions of learning and memory to decisions, one limitation of our task is that it does not provide any behavioral information about why working memory systems fail. For example, higher working memory decay in our model could reflect failures to appropriately prioritize storage of task-relevant information, including susceptibility to interference effects such as have previously been proposed to play a major role in cognitive aging (*Amer et al., 2022*), but could also be explained by a gradually increasing failure to retrieve previously stored information. Thus, although our computational approach was able to show that much of the age-related decision-making impairment previously attributed to learning is actually attributable to failures of working memory, we rely on future work to better understand the exact computational nature of those failures.

Our conclusion that age-related performance deficits were attributable to WM rather than RL is in conflict with previous literature (*Frank and Kong, 2008*; *Hämmerer et al., 2011*) suggesting age-related impairment of RL systems. Previous work on aging and RL, however, has yielded mixed conclusions, with some results suggesting that RL is not impaired in older adults (*Grogan et al., 2019*; *Radulescu et al., 2016*). Placing our study in the context of the existing literature, we believe that inconsistent results might have occurred due to a failure to separate and account for additional

processes (WM, attention, etc.) that might contribute to performance deficits in older populations. This in particular reflects our motivation for conducting this study as quantifying contribution of cognitive processes to performance should not be studied in isolation (*Radulescu et al., 2016*; *Radulescu et al., 2019*; *Collins and Frank, 2012*; *Collins, 2018*).

On the biological side, our results suggest that age-related differences in working memory, but no other age-related behavioral changes, were associated with reductions in levels of glutamate in prefrontal cortex. Previous work has noted structural changes in the aging brain, including reduced brain volume and specific reductions in gray matter in certain brain regions, including prefrontal cortex (*West, 1996*; *Buckner et al., 2004*). While we did note some structural differences between younger and older participants, these structural changes were not related to decision-making, and instead the best neural predictions of task performance relied primarily on neurochemical levels, in particular, glutmate. Glutamate is the primary excitatory neurotransmitter in the brain, and in prefrontal cortex, is thought to support local recurrent signaling that supports the active maintenance of information in cortical neural networks. An enticing interpretation of these results is that our noninvasive MRS measures, which provide regional quantification of glutamate, may provide a readout of changes in the local synaptic architecture (e.g. decreased density of excitatory synapses) that play a causal role in the cognitive aging process.

Nonetheless, our study is limited in the support that it can provide for this idea. We found a relationship between glutamate and the working memory decay parameter that explained age-related differences in task performance. However, the bulk of this relationship was driven by the group differences in levels of glutamate and working memory decay, with only a trend-level correlation between the factors existing in the older adult group. Furthermore, formal mediation modeling was unable to confirm that glutamate mediates the observed relationships between age and working memory decay. There are several potential explanations for the remaining causal ambiguity, one being that our study was not particularly well powered to see within group relationships between biological variables and behavior (24 older participants with both brain and behavioral measurements), making the observed correlation in older adults ($r = -0.39$) difficult to interpret one way or the other. Mediation analysis typically requires very large samples to ensure reasonable power levels (*Fritz and Mackinnon, 2007*) and has an additional obstacle in our case that we cannot measure our hypothesized mediating factor directly and instead rely on a noisy proxy measurement (MRS glutamate), potentially muddying the interpretation of mediation analyses (*Westfall and Yarkoni, 2016*). An alternative explanation would be that age really is the causal factor driving both metabolic changes in glutamate and, separately, cognitive declines in working memory that affect performance on learning tasks. While we are unable to conclusively arbitrate between these possibilities in this study, we hope that our results inspire future work with larger samples of older adults that can effectively do so.

Our study is also limited by the MRS methodology that we employed. A major limitation of MRS is that we are unable to verify that the age-related differences in glutamate that we measure are localized to synapses, or even to neurons. The chemical specificity of our findings lends some support to the idea that glutamate is playing a functional role rather than simply serving as one of many markers for the metabolic changes that occur over healthy aging, yet additional work leveraging animal models would be necessary to verify this idea and fully elucidate a causal mechanism. Beyond specificity, our MRS methodology limited the breadth of neurochemical measurements we could make. We did not measure levels of dopamine (DA), which is a critical biological marker of learning by reinforcement (*Schultz et al., 1997*), and could further inform our understanding of the age-related changes in neurochemistry, and its effect on cognition (*Bäckman et al., 2006*; *Li et al., 2001*). Furthermore, dopamine also might modulate WM in addition to RL (*O'Reilly and Frank, 2006*). Thus, the role of dopamine in RL-WM interaction in the aging population would be an interesting topic to explore in future research.

A third limitation of this study is the cross-sectional design. Specifically, we compared a group of young and older adults, without directly observing age-related changes within subjects (i.e. a longitudinal design). As such, the observed age group differences in performance could have been, in part, attributed to cohort effects (e.g. young adults being more comfortable with technology and computer games compared to old adults). Despite this theoretical limitation, several aspects of our data suggest that the differences we observed are related to age rather than other factors that differed across our young and older cohorts. First, we collected RBANS scores, which revealed that our two groups

reflected similar samples of their age groups with respect to overall cognitive function. Second, task performance decreased with age within the older adult group rather than simply across the two groups.

To analyze our results, we developed a Bayesian hierarchical RL-WM model that captured qualitative trends in both training and testing performance as well as group differences. However, our model validation revealed that the RL-WM model predictions deviated from older participants' data in set size 6 learning (*Figure 2B*). Further inspection revealed that this occurred because in some set size 6 blocks some older participants learned a correct response for only a subset of stimuli, while entirely neglecting to learn correct responses for the remaining stimuli in the set. One interpretation is that older adults might have used this as a strategy to minimize the difficulty of the high set size condition (ignoring some stimuli to effectively make it a smaller set size), leading participants to deviate from our model, which learned from all stimuli. This interpretation was supported by model predictions for older adults who did not exhibit such selective learning, which displayed reduced model misfit (see *Appendix 1—figure 4*). Thus, our hierarchical modeling not only captured core trends in behavior, but it also highlighted a previously undocumented discrepancy between RL-WM model predictions and learning behavior in specific individuals due to alternative strategies (e.g. neglecting stimuli). While we were unable to further characterize or quantify these strategies in our dataset, we hope that this observation inspires future work to design tasks better positioned to do so.

We have focused on testing healthy older adults in order to isolate the aging as dimension along which we tested the differences in RL and WM. In the future, it would also be valuable to examine these changes in clinical populations (e.g. individuals with mild cognitive impairment or individuals with Alzheimer's disease/dementia). Testing how neural mechanisms in these neurodegenerative disorders relate to RL/WM would further expand our understanding of the underpinnings of dementia-related symptoms. Furthermore, recognizing the signs of specific impairments at different stage of dementia development could inform cognitive-training programs that have been used to mitigate/delay cognitive symptoms in these neurodegenerative disorders (*Clare et al., 2003*). One interesting related question is that of the dynamics of age-related differences, in particular as to whether the measured age-related neural/cognitive changes are gradual or characterized by a sudden onset at a critical time point in lifespan? Based on the absence of age-related correlation with performance in young adults and negative correlation with age within the old group, it seems that the relationship between age and cognitive functions might not be linear and instead begin to decline more rapidly at older ages. Future work employing longitudinal studies could address this question and potentially link nonlinear declines in neural and behavioral measures to protracted onset of age-related disease.

In summary, we show that age-related differences in the ability to make decisions from experience are reflected in the age group variability in working memory decay. These behavioral deficits are accompanied by changes in neurochemistry, most notably reductions in prefrontal glutamate levels, that map specifically onto computational markers for working memory impairment and tend to increase with age. These findings suggest that a major component of age-related differences in decision-making could result from old adults experiencing reductions in prefrontal excitatory signaling that impair working memory systems necessary to translate recent information into appropriate action.

## Methods
### Participants
We recruited 78 participants in total (42 older adults – age mean [SD] = 68 [8.5]; 36 young adults, age mean [SD] = 21 [4.4]) for a two-session study. Older adults were recruited from the community. Young adults were recruited from the Brown University participant pool or from the community. Recruitment targeted young adults (18–30) and older adults (60–80); however, the inclusion criterion was broader and only required participants to be at least 18 y of age. Visual exclusion criteria for enrollment in the study included (1) direct report of color blindness or poor performance on Ishihara color plates, (2) a best-corrected visual acuity less than 20/40 in both eyes at distance or near, and (3) abnormalities in peripheral vision determined by confrontation visual field testing. Additional exclusion criteria related to the safety of MRI imaging excluded participants with contraindication to MRI including claustrophobia, pregnancy, or metal implants.

During the first session, all participants completed an RL-WM behavioral task as well as additional cognitive testing using the RBANS. We consider the RBANS to be a fitting choice for this as, to our knowledge, compared to alternative measures such as MMSE or dementia rating scale, the RBANS does not have ceiling effect issues and is designed to increase sensitivity to small cognitive differences. During the second session, participants underwent MRS and completed additional behavioral testing. Out of 36 young adults, 6 did not complete the MRS session; out of 42 old adults, 18 did not complete the MRS session. These participants were omitted from the analyses that linked behavioral/modeling data and neural measures. The two sessions were separated by a maximum of 7 d. Participants received monetary compensation for participating in the study. All participants provided a written informed consent prior to beginning the experiment. All procedures were approved by the Brown University Institutional Review Board under protocol #0812992595 (behavioral session) and #1203000583 (MRS session).

## Task

A general format of reinforcement learning experiments assesses subjects' engagement of the feedback-dependent learning process, which enables them to store rewarding stimulus–response associations. The RL-WM task employs the same logic, while simultaneously manipulating the working memory (WM) involvement by varying the number of stimulus–response associations participants are required to learn in a given block (*Collins and Frank, 2012*; *Collins, 2018*; *Collins and Frank, 2018*; *Figure 1*). In the previous work, this manipulation yielded a way to disentangle the contribution of WM and RL to the learning process (*Master et al., 2020*).

Given that WM is capacity limited and susceptible to decay, participants are more likely to have stored in working memory an informative stimulus–action association for a given trial when there are fewer items to store. Since WM enables fast, albeit short-lasting and limited, storage, we expected the accuracy to asymptote within very few trials – if the number of associations is within the capacity bounds. On the other hand, if the number of associations participants are required to store exceeds their WM capacity, they would be more likely need to use a more incremental but more robust system to make a choice (RL). Therefore, having two conditions (small and high set size) enabled us to dissociate the contribution of these two learning systems.

### Learning phase

Before starting the actual task, participants received detailed task instructions and a brief practice phase. They were told that the goal of the experiment was to learn a correct key press in response to the given images in order to earn as many points as possible. We provided participants with trial examples during which they were presented with a stimulus on the screen – they had 1 s to press one of the three keys and were subsequently given a 1 s deterministic feedback. If they selected a correct response, participants received a point (+1); if they selected a wrong response key, they received 0 points. Participants advanced to the next trial following the feedback termination (*Figure 1A*). Points were translated to an incentive payment at the end of the study.

Participants learned in 10 independent blocks, 6 with small set size (three stimuli per block) and 4 with high set size (six stimuli per block), each block with a novel set of images. There were more small-set size blocks due to the fact that learning fewer stimulus–response associations provides a noisier assessment of the learning performance. In each block, each stimulus had a correct associated action that the participant needed to discover through trial and error (e.g. action 3 is correct for stimulus 1) and appeared nine times during the block. The stimulus order was pseudo-randomized to ensure a uniform distribution of delay between two successive presentations of the same stimulus (*Collins, 2018*). We counterbalanced correct stimulus–response mappings across the blocks.

### Testing phase

After the training phase, participants completed an unrelated 20 min task (probabilistic selection task [PST]; *Frank et al., 2004*) that was visually distinct from the RL-WM task and served to introduce an extended delay between training and the test phase. After completing the PST task, participants were again tested on their knowledge of the stimulus–response associations learned in the RL-WM training blocks. However, unlike in the training phase, feedback was omitted to prevent new learning. Therefore, participants could only be informed by feedback received at least 20 min prior to the test

phase. Thus, compared to the learning phase where WM highly contributes to performance, the test phase reflects better isolated slow, long-term learning processes (RL). Specifically, due to (1) a 20 min delay and (2) omitted feedback, WM memory involvement in test phase is effectively eliminated since WM cannot be engaged in new learning (due to absence of feedback), and the time delay over which information stored in WM has decayed. Investigating test phase performance in comparison to learning phase performance allows us to see how learning within the RL process is impacted by the use of WM during learning. If our prediction is correct, we should observe a larger decrease from training to testing performance in set size 3 (which could be learned via WM) compared to set size 6 condition (that exceeds the WM capacity and is more likely to be learned via RL that is more robust across increased delay intervals). In our task (and model), we assume that the test phase performance is supported solely by RL in the absence of WM. Nevertheless, we acknowledge that it could also reflect the contribution of an alternative long-term storage system, such as episodic memory (**Bornstein and Norman, 2017**), but this task is not designed to identify it.

## Computational model

In order to understand the cognitive factors that drive age-related differences in behavior, we used a hybrid reinforcement learning (RL) and working memory (WM) model (RL-WM). This model is designed to disentangle the contributions of RL and WM to learning by accounting for each process in a separate RL and WM module. The RL module tracks the *values*, $Q$, of stimulus–action associations, which are learned incrementally from the reward history. Specifically, the RL module in the model means to capture incremental learning that is not capacity limited, analogous to the RL thought to be implemented by dopaminergic processes in the brain (**Schultz et al., 1997**). The WM module acquires stimulus–action association *weights*, $W$, through fast, one-shot learning, but its stored associations decay over time. By modeling learning within the two modules and their weighted contribution to action selection, the RL-WM model is able to quantify the relative influence of RL and WM processes on learning (**Collins and Frank, 2012**; **Collins, 2018**; **Collins et al., 2014**).

### RL learning rule

The incremental learning of stimulus–action values in the RL module is based on a simple delta rule (**Sutton and Barto, 2018**). Specifically, on each trial $t$, the value $Q(s, a)$ of the action $a$ made in response to the presented stimulus $s$ is updated in proportion to the reward prediction error $\delta$ (difference between expected and observed outcome):

$$\delta_{\mathrm{RL}} = r - Q_t(s, a)$$

$$Q_{t+1}(s, a) = \begin{cases} Q_t(s, a) + \alpha^+ \, \delta_{\mathrm{RL}} & \text{if } \delta_{\mathrm{RL}} > 0 \\ Q_t(s, a) + \alpha^- \, \delta_{\mathrm{RL}} & \text{if } \delta_{\mathrm{RL}} \leq 0 \end{cases}$$

where $\alpha^+$ and $\alpha^-$ are positive and negative learning rates, and $r$ is the outcome for incorrect and correct trials (0 or 1 point, respectively). Previous work suggests that individuals learn differently from positive and negative feedback, specifically suggesting that they are more likely to neglect negative feedback when learning rewarding responses (**Frank et al., 2007**; **Gershman, 2015**; **Niv et al., 2012**). To address this property of learning, we allow for separate learning rates when updating $Q$-values, depending on the sign of the prediction error.

$Q$-values are initialized at $Q_0 = 1/n_A$ (where $n_A$ is the number of possible response actions) at the start of each block. These are uniform values (equal values for all S-A associations) that reflect the reward expectation in the absence of learned information.

### WM learning rule

In contrast, the WM module is a one-shot learning system that immediately stores and retains the information from the previous trial. To model this, we quantified WM stimulus–action weights as storing the immediate outcome of the trial:

$$W_{t+1}(s, a) = r \qquad\qquad \text{if } \delta_{\mathrm{WM}} > 0$$

The neglect of negative feedback is also assumed to affect WM. To capture this, we allow for imperfect encoding of the outcome as an association weight when the WM module's prediction error is negative. The strength of this imperfection $\nu$ is identical to the relative neglect of negative feedback in the RL module, $\frac{\alpha^-}{\alpha^+}$.

$$W_{t+1}(s,a) = W_t(s,a) + \nu(r - W_t(s,a)) \qquad \text{if } \delta_{\text{WM}} \leq 0$$

However, we note that as the positive and negative learning rates are not subject to any order constraint, $\nu$ is permitted to be greater than 1, which would imply a relative preference to learn from negative feedback.

The WM weights are initialized to $W_0$ that is defined similarly to the initial $Q$-values, but unlike the $Q$-values, WM weights are susceptible to decay $\phi$. On each trial, the $\phi$ parameter pulls the WM weights to their initial values $W_0$:

$$W_{t+1}(s_i,a_j) = W_t(s_i,a_j) + \phi(W_0(s_i,a_j) - W_t(s_i,a_j)) \qquad \forall s_i \, \forall a_j \neq (s,a)$$

This decay applies to all stimulus–action associations except the exact association seen in the current trial.

## Policy

Both the RL and WM modules contribute to the likelihood on each trial of choosing each of the $n_A$ possible actions. To generate an action policy within each module, we transform the $Q$-values and WM weights separately into choice probabilities:

$$\begin{aligned} P_{\text{RL}}(a|s) &= \frac{\exp(\beta^{\text{L}} \, Q_t(s,a))}{\sum_{i=1}^{n_A} \exp(\beta^{\text{L}} \, Q_t(s,a_i))} \\ P_{\text{WM}}(a|s) &= \frac{\exp(\beta^{\text{L}} \, W_t(s,a))}{\sum_{i=1}^{n_A} \exp(\beta^{\text{L}} \, W_t(s,a_i))} \end{aligned} \qquad (1)$$

These softmax policies imply that actions with higher $Q$-values and WM weights will be selected with higher probability. The inverse temperature $\beta^{\text{L}}$, which applies to both softmax functions, controls the overall extent to which the overall choice process is deterministic during the learning phase. Higher $\beta^{\text{L}}$ values imply that the process will be more deterministic and less exploratory.

To integrate the RL and WM modules' policies, the RL-WM model assumes that the choice is generated as a function of a weighted mixture of the RL and WM policies, where this proportional weighting is determined by a WM weight $\omega$ parameter that quantifies one's relative reliance on WM:

$$P_{\text{RL-WM}}(a|s) = \omega^{n_S} \, P_{\text{WM}}(a|s) + (1 - \omega^{n_S}) \, P_{\text{RL}}(a|s)$$

As the number of associations to store exceeds WM's capacity, individuals should rely less on WM and more on RL. To capture this, we allow the relative reliance on each process to depend on the set size $n_S$. Higher $\omega^{n_S}$ values imply greater reliance on WM and lower values imply greater reliance on RL while learning $n_S$ stimulus–action associations. Modeling weighted contribution of RL and WM to policy allows us to capture the effect of learning within each module on action selection, and thus disentangle the contribution of these two modules to performance.

We further extended the policy to capture random lapses in the choice process. Specifically, individuals often make value-independent, random lapses in action – independent of the learning process. To capture this behavioral property, we added a random noise parameter in choice selection in the final policy (***Collins and Frank, 2012***; ***Nassar and Frank, 2016***):

$$P = (1 - \epsilon) \, P_{\text{RLWM}} + \epsilon \, \frac{1}{n_A} \qquad (2)$$

where $1/n_A$ is the uniform random policy, and $\epsilon$ is the noise parameter.

## Test phase

Given that the test phase is administered with a delay (thus eliminating WM contribution), we assume that the choice process during this phase is exclusively supported by RL (see 'Task description'). The

choice policy is based only on the $Q$-values learned for the stimulus–action associations at the end of each learning block:

$$P_{\text{RL}}^{\text{test}}(a|s) = \frac{\exp(\beta^{\text{T}} Q_t(s, a))}{\sum_{i=1}^{n_A} \exp(\beta^{\text{T}} Q_t(s, a_i))}$$

$$P^{\text{test}} = (1 - \epsilon) P_{\text{RL}}^{\text{test}} + \epsilon \frac{1}{n_A}$$

Here, $\beta^{\text{T}}$ is the inverse temperature specific to the test phase (i.e. different from the learning phase $\beta^{\text{L}}$). We also assumed some noise in this decision process, with the same lapse rate $\epsilon$ as in the learning phase (see *Equation 2*). $Q$-values are no longer updated as there is no feedback given during the test phase.

The list of free parameters for the RL-WM model includes a positive learning rate ($\alpha^+$) and a negative learning rate ($\alpha^-$), two inverse temperatures (in each phase of the task, $\beta^{\text{L}}$ and $\beta^{\text{T}}$), random lapse rate ($\epsilon$), decay rate ($\phi$), and two degrees of reliance on WM (in each set size condition, $\omega^3$ and $\omega^6$). For every participant, we inferred the value of each of these parameters except $\beta^{\text{L}}$, for which we made the simplifying assumption that a singular value applies to all participants in the same age group. Most recent work with the RL-WM model has fixed $\beta^{\text{L}}$ to the a singular value (often 50 or 100) for all participants (*Collins, 2018*; *Master et al., 2020*); our approach is similar, but ultimately more flexible as it effectively allows the data to dictate what value to fix for $\beta^{\text{L}}$.

A schematic representation of how the RL-WM process generates behavioral data in both the learning and testing phases of the RL-WM task is presented in *Figure 2A*.

## Hierarchical Bayesian formulation of the RL-WM model

To estimate the parameters of the RL-WM model for all participants simultaneously, we developed the first hierarchical Bayesian formulation of the RL-WM model. This Bayesian implementation is a novel extension of the RL-WM model, allowing for the inclusion of domain knowledge directly in the model (via carefully developed priors) and the incorporation of theoretically meaningful hierarchical structure. A hierarchical Bayesian approach to model fitting offers many benefits, including the ability to quantify the uncertainty in each parameter estimate and the regularization of extreme estimates with respect to the group (for an introduction, see *Lee, 2011*).

We compared three versions of the Bayesian RL-WM model with different hierarchical structures. First, we defined a non-hierarchical model by specifying a prior for each unobserved parameter of the RL-WM model's data-generating process (as described above). This would be equivalent to fitting the model separately to each participant's data, were it not for our simplifying assumption that the same value of $\beta^{\text{L}}$ is used for all participants. Next, we considered a hierarchical extension of the model over participants, such that all $P$ participants' parameter values (e.g. all 78 learning rates, $\alpha_1, \alpha_2, \ldots, \alpha_P$) are assumed to be drawn from a group-level distribution (e.g. $\alpha_p \sim \text{Beta}(1 + a, 1 + b) \,\forall p$). Finally, we considered a model with separate hierarchies over the participants within each age group. This version of the RL-WM includes different hyperparameters for each age group (e.g. $a_{\text{young}}$, $b_{\text{young}}$ vs. $a_{\text{older}}$, $b_{\text{older}}$), which would allow us to make inferences about group-level differences in each dynamic of the RL-WM process.

For each version of the model, we specified priors for participant-level parameters and hyperpriors for group-level hyperparameters that were mildly to moderately informative. The priors and hyperpriors used to specify the winning two-group model are detailed in Appendix 1. Prior predictive checks and other simulation-based procedures were used to confirm that the set of prior distributions and the RL-WM model likelihood taken together specified a reasonable model (*Kennedy et al., 2019*; *Baribault and Collins, 2023*; see *Appendix 1—figure 1*).

## Model fitting

For each of the three candidate models, we used Stan (*Carpenter et al., 2017*) to estimate the joint posterior distribution of all model parameters via Markov chain Monte Carlo (MCMC) sampling. After running a model with four chains of 500 warmup iterations and 1500 kept iterations each, we performed a series of diagnostic checks in accordance with current best practices for MCMC methods (*Vehtari et al., 2021*; *Betancourt, 2016*; *Gelman et al., 2013*). We required an $\hat{R}$ value of $\leq 1.01$ and an effective sample size of $\geq 400$ for all parameters, a BFMI of $\geq 0.2$ for all chains, and that no

divergences were observed. Only kept iterations from model output that met these criteria were used for analysis.

We used the (widely applicable or Watanabe–Akaike information criterion [WAIC]; *Watanabe, 2010*) as our model comparison metric as it is fully Bayesian, invariant to parameterization, and straightforward to compute. Because response data from successive trials of the RL-WM task are not independent, we consider one block of data from one participant as the smallest unit of data when computing log likelihood for WAIC. As WAIC is an estimate of out-of-sample prediction error, lower WAIC values indicate a better model. Our final model selection was further supported by posterior predictive checks of the models' relative descriptive adequacy (see *Appendix 1—figure 2*). In the model-based analyses reported here, all intervals are 95% equal-tailed credible intervals (CrI) unless otherwise specified.

## MRS procedure

Glutamate and GABA levels were measured in three brain regions (striatum [STR], middle frontal gyrus [MFG], and intraparietal sulcus [IPS]) using MRS. First, structural images of the participant brain were acquired using a high-resolution T1-weighted sequence (MPRAGE, TR/TE/TI = 1900/3.02/900ms, 9° flip angle, $1.0 \times 1.0 \times 1.0$ mm³ voxel resolution) with a 64-channel RF receive coil array on a 3T MRI scanner (Siemens MAGNETOM Prisma). Next, we manually placed voxel bounding boxes in each brain region for MRS data acquisitions. The striatal voxel was $18 \times 24 \times 15$ mm and was placed such that the anterior portion of the voxel started at the anterior extreme of the head of the caudate and that it extended posterior to maximize the inclusion of caudate and putamen and minimize inclusion of ventricular space. A 20 mm cubic voxel was placed in the right MFG immediately anterior to the precentral sulcus and immediately inferior to the superior frontal sulcus. A 20 mm cubic voxel was placed to be centered at the anterior-ventral section of the left IPS. Representative images of voxel placements are shown in *Figure 3*.

For each voxel, we acquired measures of GABA using a MEGA-PRESS sequence (*Hu et al., 2013*) (TR = 1500 ms, TE = 68 ms, average = 192) with double-banded pulses, which were used to simultaneously suppress water signal and edit the γ-CH2 resonance of GABA at 3 ppm. Linear and second-order shimming was used to achieve typical linewidth (~14 Hz). Difference spectra were obtained by subtracting the signals obtained from the selective double-banded pulse applied at 1.9 and 4.7 ppm ('Edit on') from those obtained from the double-banded pulse applied at 4.7 and 7.5 ppm ('Edit off'). Glutamate scans were conducted using the PRESS sequence (*Hancu, 2009*) (TR = 3000 ms, TE = 30 ms, 90° flip angle, average = 64). A variable pulse power and optimized relaxation delays (VAPOR) technique was used in both sequences to achieve water suppression (*Tkác et al., 1999*). While MRS data were collected, participants completed a simple task that required them to push a button whenever a fixation point changed color (*Shibata et al., 2017*).

Glutamate was quantified using LC-model (*Provencher, 2001*) that attempts to fit the average signal in the frequency domain using a linear combination of basis functions. Separate basis functions were used to quantify glutamate and glutamine, a molecule that has similar spectral characteristics to glutamate in the PRESS sequence. The reliability of glutamate quantifications was indicated by the Cramer–Rao lower bounds, and a criterion of 20% was chosen to reject low-quality signal; however, no measurements were rejected due to this criterion.

GABA was quantified using a custom peak integration process implemented in MATLAB. This quantification procedure was used because it produced higher split half reliability on our dataset than standard GABA quantification software (*Edden et al., 2014*). Raw data were obtained from the Siemens Prisma scanner in the form of the so-called 'twix' file that contains the (complex) individual signals from each receive channel for every average. Signals underwent phase shifting to equalize the starting phase of all of the receive channel signals prior to combining for each signal average. Signals were combined using weighting factors based on individual signal amplitudes. Starting with the highest amplitude signal, channels were added until there was no further increase in NAA peak SNR. Free induction decays were averaged separately for 'off' and 'on' pulse conditions and subtracted to produce a difference signal. The difference signal was transformed into frequency space with a fast Fourier transform and frequencies were converted to parts per million, where 1 PPM is equal to 123.255 MHz, and relative to a water reference of 4.8 PPM. The GABA peak was quantified by integrating chemical shifts ranging from 2.9 to 3.1 ppm and subtracting out the integrated signal in

a surrounding reference window (2.8–2.9, 3.1–3.2 ppm). This procedure was validated through split half correlations in which the entire procedure was performed separately for odd and even acquisition volumes, and resulting GABA measurements for odd and even acquisitions were correlated with one another to yield relatively high reliability for MFG and IPS measures ($R = 0.80$ for both regions) and moderate reliability for striatal GABA measures ($R = 0.68$; see *Appendix 1—figure 6*).

Freesurfer 6.0 was used to reconstruct T1 anatomical scans from DICOM images collected using the MPRAGE sequence following the recon-all -all pipeline. The initial pial surface outputs were manually corrected for brainmasks that included skull tissues as described in the Freesurfer quality control documentation (https://freesurfer.net/fswiki). Next, partial volume fractions for gray matter (cortex + subcortical), white matter, and cerebrospinal fluid were computed as voxel maps using the mri_compute_volume_fractions Freesurfer command based on the edited T1 reconstruction. Then, masks for the three brain regions (striatum, MFG, and IPS) measured with MRS were constructed using the Gannet 2.1 toolbox (*Edden et al., 2014*) based on the voxel bounding boxes manually placed during the MRS acquisition. Gray/white matter volumes in each region were computed by summing partial volume voxel maps with masking. Similarly, anatomical measures within each region were obtained by summing tissue maps with masking using the mri_segstats Freesurfer command on labelled parcellation of the cortex using the Desikan–Killiany atlas (*Desikan et al., 2006*) and automatic segmentation of the subcortical structures (*Fischl et al., 2002*).

## Behavioral analyses

To visualize participants' learning trajectory, we created learning curves by collapsing participants' accuracy at each stimulus iteration. This enabled us to separate early learning and asymptotic performance (late learning accuracy). Trials with missing responses (i.e. where participants did not respond within the time limit) were excluded from analyses.

While learning curves are useful for visualizing learning trajectory, they cannot be used to make inferences about trial-by-trial contribution of different factors (i.e. related to RL or WM) to participants' accuracy. Thus, to quantify performance in terms of factors related to RL and WM, we ran a trial-by-trial analysis using a mixed-effects general linear model (GLM), predicting accuracy (coded as 0 or 1 on each trial). Specifically, the predictors in the logistic regression consisted of the following: set size (the number of associations), delay (number of intermediate trials between two successive rewarded stimulus representations), and reward history (stimulus-dependent cumulative reward history). We also added trial and block number predictors to control for overall improvement across the task. We used participants' coefficients to predict their respective age using a linear model in order to draw a relationship between age and RL and WM-related factors' effect on performance. We excluded coefficients from two participants as these coefficients were over 2 SD above the mean following the approach implemented in previous work (*Master et al., 2020*).

## Data-driven approach to identifying best MRS predictors

To examine the effect of brain measures on the parameters and task performance, we first had to reduce the dimensionality of highly correlated MRS data by identifying the best set of MRS predictors in explaining participants' learning performance. We focused on learning performance because unlike the test phase the learning phase contains signatures of both WM and RL (see 'Task description).

First, we averaged all measures that were sampled from multiple regions across those regions (e.g. glutamate predictor in the model was an average of glutamate measures across three regions it was sampled from). The only measure that was not averaged was cortical thickness (from CMF, RMF and, SF) because these three regions were selected a priori from a Freesurfer parcellation (see 'MRS methods') and are unrelated to the voxels. We averaged the predictors in order to reduce the number of predictors in the model in a meaningful way (e.g. instead of three different glutamate features we would have only one) and because our power would have suffered massively due to collinearity in our predictors. As a result, we used the following features to construct a GLM: gray matter, white matter, and cortical thickness in SF, CMF, and RMF cortex (structural measures), glutamate, GABA, and NAA (neurochemical measures). Next, we implemented a $k$-fold cross-validation approach (a commonly used method in machine learning) to identify the best configuration of predictors that minimizes the loss (mean squared error) in out-of-sample prediction. Specifically, we randomly sampled 4/5 of the data to train the model, leaving out the remaining 1/5 of the data for validation, and repeated this

process 200 times. On each of the 200 iterations, we stored the best configuration of predictors that minimized the loss of the model's prediction of the out-of-sample performance. We also added the regularization term, which represented a combination of $L^1$ and $L^2$ penalty (i.e. elastic net; with alpha = 0.5 such that ridge ($L^2$) and lasso ($L^1$) optimization are weighted equally; regularization coefficient $\lambda$ was set to default – lasso sets the maximum value of $\lambda$ that gives a non-null model) to help us eliminate the predictors that did not significantly contribute to performance and might lead to overfitting. We reasoned that this data-driven approach was a principled way of analyzing highly colinear measures that would allow us to reduce the dimensionality of our data as well as the number of hypothesis tests required to draw conclusions. After our machine learning model comparison, we fit our winning neural model using a standard regression and performed significance testing on individual beta estimates to identify the neural measures that were significantly associated with training performance within the best model.

Once we obtained the best set of features based on the cross-validation approach, we used this model to generate the brain-based performance prediction. The brain-based predicted performance represented a mapping between neural measures and behavior – that is to say, based on our brain measures, what level of training performance might be expected? Next, we regressed brain-based performance predictions onto the fit parameters from our RL-WM model in order to test which parameters were most closely related to the dimension of performance that we could identify using neural measures. This step also allowed us to measure the relationships between model parameters (i.e. decay, learning rate) and the brain-based performance predictions while controlling for all other model parameters, thereby improving the specificity with which we could map neural correlates of performance onto specific computational processes. Significance testing was performed to identify which parameters were associated with brain-based performance measures.

Follow-up analyses were conducted on neural measures with significant associations to performance and model parameters with significant associations with brain-based performance predictions. Each model parameter with a significant relationship was regressed onto anatomically specific neurochemical measures that were significantly associated with performance. In practice, this follow-up analysis was only applied to the decay and set size 3 WM weight parameters, which were regressed onto glutamate concentrations in MFG, IPS, and STR.

## Acknowledgements

This work was supported by the National Institute on Aging grants K99AG054732 and R00AG054732 to MRN, and NSF 2020844 to AGEC. We thank Rachel Rac-Lubashevsky, Michael Frank, and Aaron Fisher for helpful comments.

## Additional information

### Funding

| Funder | Grant reference number | Author |
| --- | --- | --- |
| National Institute on Aging | R00AG054732 | Matthew R Nassar |
| National Institute on Aging | K99AG054732 | Matthew R Nassar |
| National Science Foundation | NSF2020844 | Anne GE Collins |

The funders had no role in study design, data collection and interpretation, or the decision to submit the work for publication.

### Author contributions

Milena Rmus, Data curation, Formal analysis, Investigation, Visualization, Methodology, Writing - original draft, Writing - review and editing; Mingjian He, Data curation, Formal analysis, Investigation, Visualization, Methodology, Writing - original draft, Project administration, Writing - review and editing; Beth Baribault, Formal analysis, Investigation, Visualization, Methodology, Writing - original draft, Writing - review and editing; Edward G Walsh, Investigation, Methodology, Project administration,

Writing - review and editing; Elena K Festa, Supervision, Investigation, Writing - review and editing; Anne GE Collins, Supervision, Investigation, Methodology, Writing - review and editing; Matthew R Nassar, Conceptualization, Supervision, Funding acquisition, Investigation, Methodology, Writing - original draft, Project administration, Writing - review and editing

**Author ORCIDs**
Milena Rmus ⓘ http://orcid.org/0000-0002-2044-048X
Mingjian He ⓘ http://orcid.org/0000-0002-6688-8693
Elena K Festa ⓘ http://orcid.org/0000-0002-3700-4270
Anne GE Collins ⓘ http://orcid.org/0000-0003-3751-3662
Matthew R Nassar ⓘ http://orcid.org/0000-0002-5397-535X

**Ethics**
Human subjects: All participants provided a written informed consent prior to beginning the experiment. All procedures were approved by the Brown University Institutional Review Board under protocol 0812992595 (behavioral session) and 1203000583 (MRS session).

**Decision letter and Author response**
Decision letter https://doi.org/10.7554/eLife.85243.sa1
Author response https://doi.org/10.7554/eLife.85243.sa2

## Additional files

**Supplementary files**
• MDAR checklist

**Data availability**
All data and code has been made available on OSF.

The following dataset was generated:

| Author(s) | Year | Dataset title | Dataset URL | Database and Identifier |
|---|---|---|---|---|
| Milena R, Nassar M | 2023 | Age-related differences in prefrontal glutamate are associated with increased working memory decay that gives appearance of learning deficits | https://doi.org/10.17605/OSF.IO/2U7PM | Open Science Framework, 10.17605/OSF.IO/2U7PM |

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

# Appendix 1

## Model specification

The data distribution for the two-group Bayesian RL-WM model (i.e. the model used for all of our model-based analyses) is described in full detail in the main text of the article. We include all prior distributions here to complete the hierarchical Bayesian model specification.

We set the following priors on participant-level parameters:

$$\alpha_p^+ \sim \text{Beta}(1 + a_{g|p}^{\alpha^+}, 1 + b_{g|p}^{\alpha^+})$$

$$\alpha_p^- \sim \text{Beta}(1 + a_{g|p}^{\alpha^-}, 1 + b_{g|p}^{\alpha^-})$$

$$\phi_p \sim \text{Beta}(1 + a_{g|p}^{\phi}, 1 + b_{g|p}^{\phi})$$

$$\omega_p^3 \sim \text{Beta}(1 + a_{g|p}^{\omega^3}, 1 + b_{g|p}^{\omega^3})$$

$$\omega_p^6 \sim \text{Beta}(1 + a_{g|p}^{\omega^6}, 1 + b_{g|p}^{\omega^6})$$

$$\epsilon_p \sim \text{Beta}(1 + a_{g|p}^{\epsilon}, 1 + b_{g|p}^{\epsilon})$$

$$\beta_p^T \sim \text{Gamma}(1 + \alpha_{g|p}^{\beta^T}, \beta_{g|p}^{\beta^T})$$

The subscript $p$ indicates that identical priors were set $\forall\, p = 1, 2, \ldots, P$ participants. The subscript $g|p$ indicates use of the hyperparameter corresponding to the group membership of that participant (where $g = 1$ and $g = 2$ denote the young and old age groups, respectively).

We set the following prior on both $\beta^L$ parameters:

$$\beta_g^L \sim \text{Gamma}(5, 0.4)$$

The subscript $g$ indicates that this same prior was used for the $\beta^L$ parameters specific to each of the two age groups.

We set the following hyperpriors on group-level parameters: 1.3 $a_g^{\alpha^+} b_g^{\alpha^+} a_g^{\alpha^-} b_g^{\alpha^-}$ 1.3 $a_g^{\phi} b_g^{\phi} b_g^{\epsilon} b_g^{\epsilon}$ 1.3 $a_g^{\omega^3} b_g^{\omega^3} a_g^{\omega^6} b_g^{\omega^6}$ 1.3 $\alpha_g^{\beta^T} \beta_g^{\beta^T}$ The subscript $g$ indicates that this same prior was used for both age groups. For any pairs of parameters that we considered comparing directly, we ensured that these comparisons would not be biased by any potential lingering influence of the prior by specifying identical hyperpriors. Such pairs include the learning rates, $\alpha^+$ and $\alpha^-$, and the mixture parameters, $\omega^3$ and $\omega^6$.

As these hyperparameters are not readily interpretable, we derived group-level means and standard deviations for each RL-WM model parameter post hoc, sample-by-sample.

## Prior predictive checks

To ensure models as-specified were capable and reasonable as models of RL-WM task data, we performed prior predictive checks (*Baribault and Collins, 2023*). We generated prior predictive distributions by randomly drawing group-level parameter values directly from the hyperpriors using those values to draw participant-level parameter values directly from the priors and using those parameter values to simulate a dataset with the same experimental design as we used in our behavioral data collection (six $n_S = 3$ blocks and three $n_S = 6$ blocks; nine iterations per stimulus in each block and in the test phase; three possible response actions). 100 prior predictive datasets were simulated for the prior predictive checks shown here.

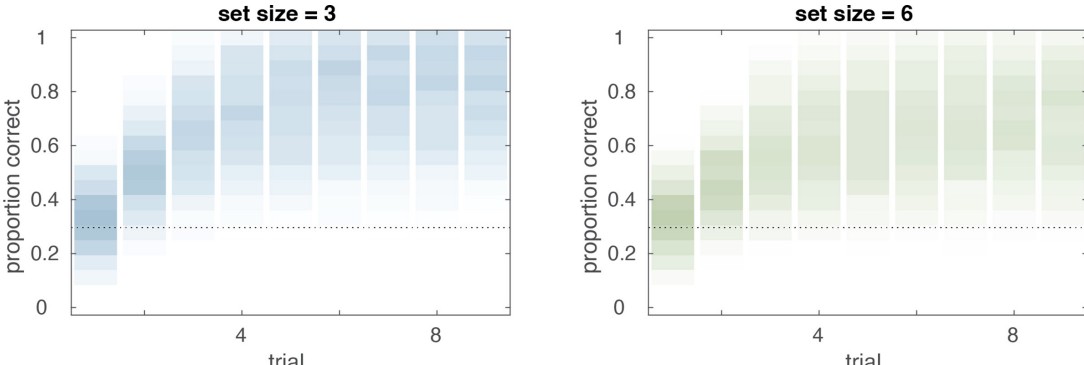

**Appendix 1—figure 1.** Prior predictive checks. Prior predictive checks for the hierarchical Bayesian reinforcement learning–working memory (RL-WM) model. The distribution of learning curves in each set size condition that are implied solely by the model specification suggests the model specification is suitable for the RL-WM task data. Group-level learning curves we would never expect to see (e.g. anti-learning) are not given any notable prior weight; while all possible group-level learning curves we could conceivably observe are given some prior predictive weight, and the most likely learning curves are not excessively strongly weighted. Furthermore, it is nice to see that the set size effect is an emergent property of our model specification.

## Posterior predictive checks

After model fitting, we performed posterior predictive checks to check the descriptive adequacy of each candidate Bayesian RL-WM model. To generate the posterior predictive distribution, we used the last 500 samples collected (i.e. the last 125 iterations from each chain). Each of these samples from the joint posterior parameter was used to generate a new dataset using an identical experiment structure (same number of participants, same stimulus sequences, etc.)

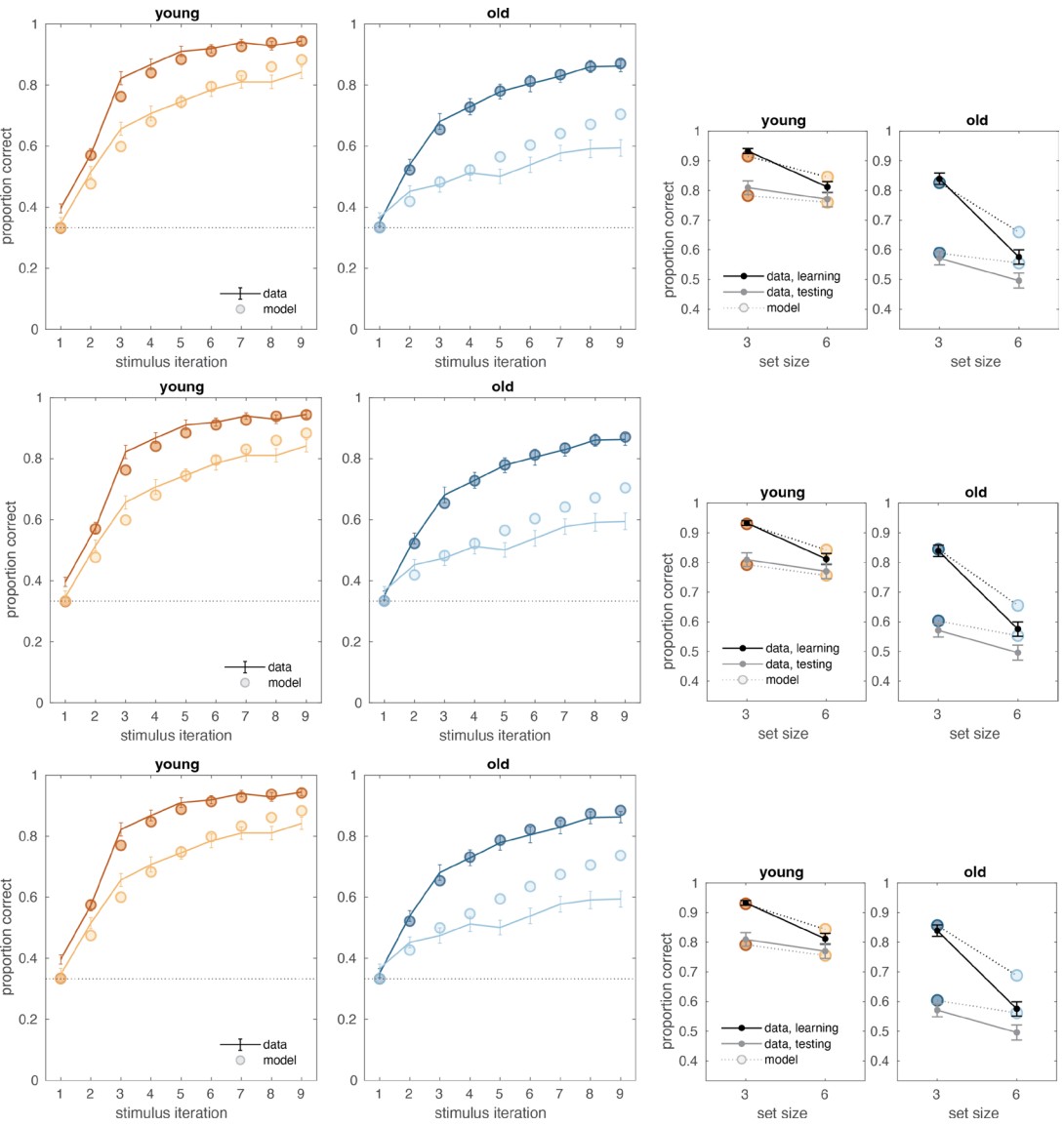

**Appendix 1—figure 2.** Posterior predictive checks. Posterior predictive checks for learning curves (left) and asymptotic means (right) for the non-hierarchical (top row), single-group (middle row), and two-group (bottom row) versions of the Bayesian reinforcement learning–working memory (RL-WM) model. (Note that all model-based analyses reported in the main text are based on output from the winning two-group version of the model.)

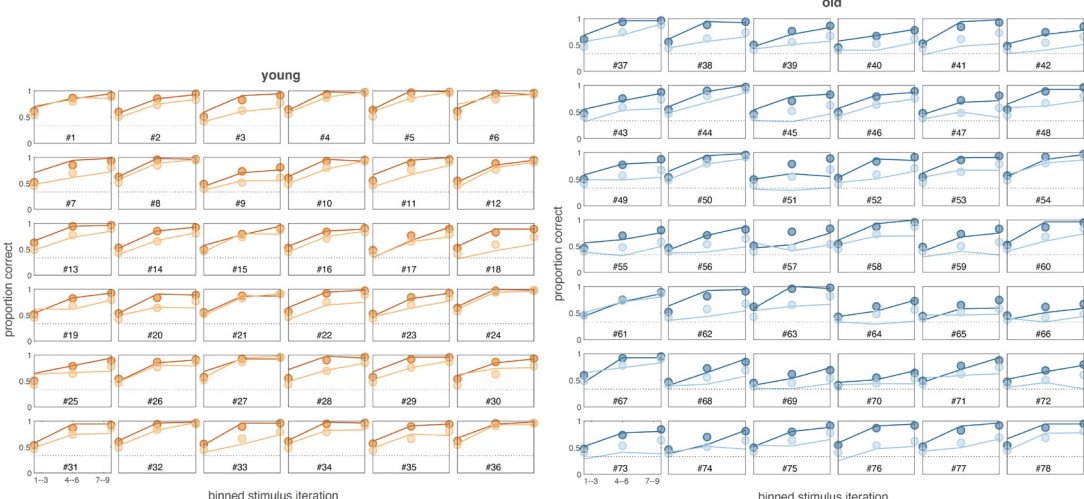

**Appendix 1—figure 3.** Posterior predictive checks of learning curves for each participant for the two-group version of the Bayesian reinforcement learning–working memory (RL-WM) model. As a smaller amount of data is available for individual participants, we plot the curve over stimulus iteration bins (of three iterations each). Our Bayesian RL-WM model captures the performance very well for nearly all participants in the young age group and for many participants in the older age group. Some participants in the older age group are not fit as well, such as #63, and a few are severely misfit, such as #51.

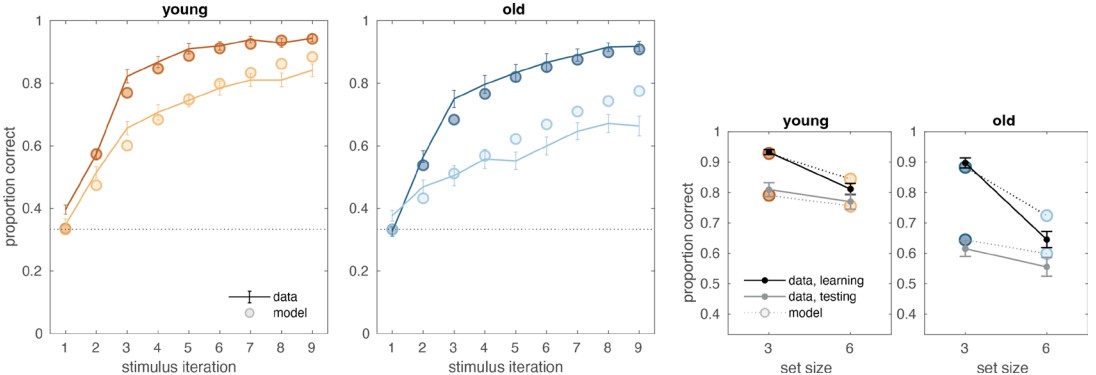

**Appendix 1—figure 4.** Posterior predictive checks for the two-group version of the Bayesian reinforcement learning–working memory (RL-WM) model when data from 14 participants was excluded. The criterion for exclusion was that mean accuracy on at least one set size 3 block was at or below chance when missed responses were classed as incorrect (which is *only* the case for this analysis).

*Appendix 1—figure 5 continued on next page*

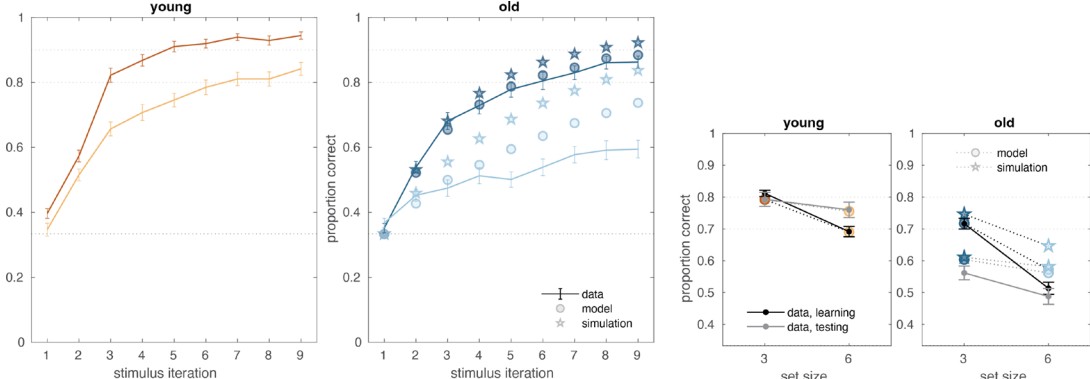

**Appendix 1—figure 5.** Exploratory demonstration of the role of decay rate $\phi$ in older adults' performance. Posterior predictive checks were recomputed after replacing the posterior samples for individual-level decay parameters for all older adults group with the posterior samples for the group-level decay rate of the young adults. While these simulation results are not sufficient as the basis for a quantitative measure or test, examining the difference in the reinforcement learning–working memory (RL-WM) model's behavioral predictions before (circles) and after (stars) this substitution is revealing: it suggests the lower decay rate might be responsible for most of the difference in behavioral performance between the age groups during the learning phase, particularly for set size 6. Decay rate can recover less of the group difference during the test phase (as the test-phase inverse temperature $\beta^T$ exerts stronger control in this portion of the RL-WM model). Lighter dotted lines are included to facilitate comparison across groups; the black dotted lines represent chance performance.

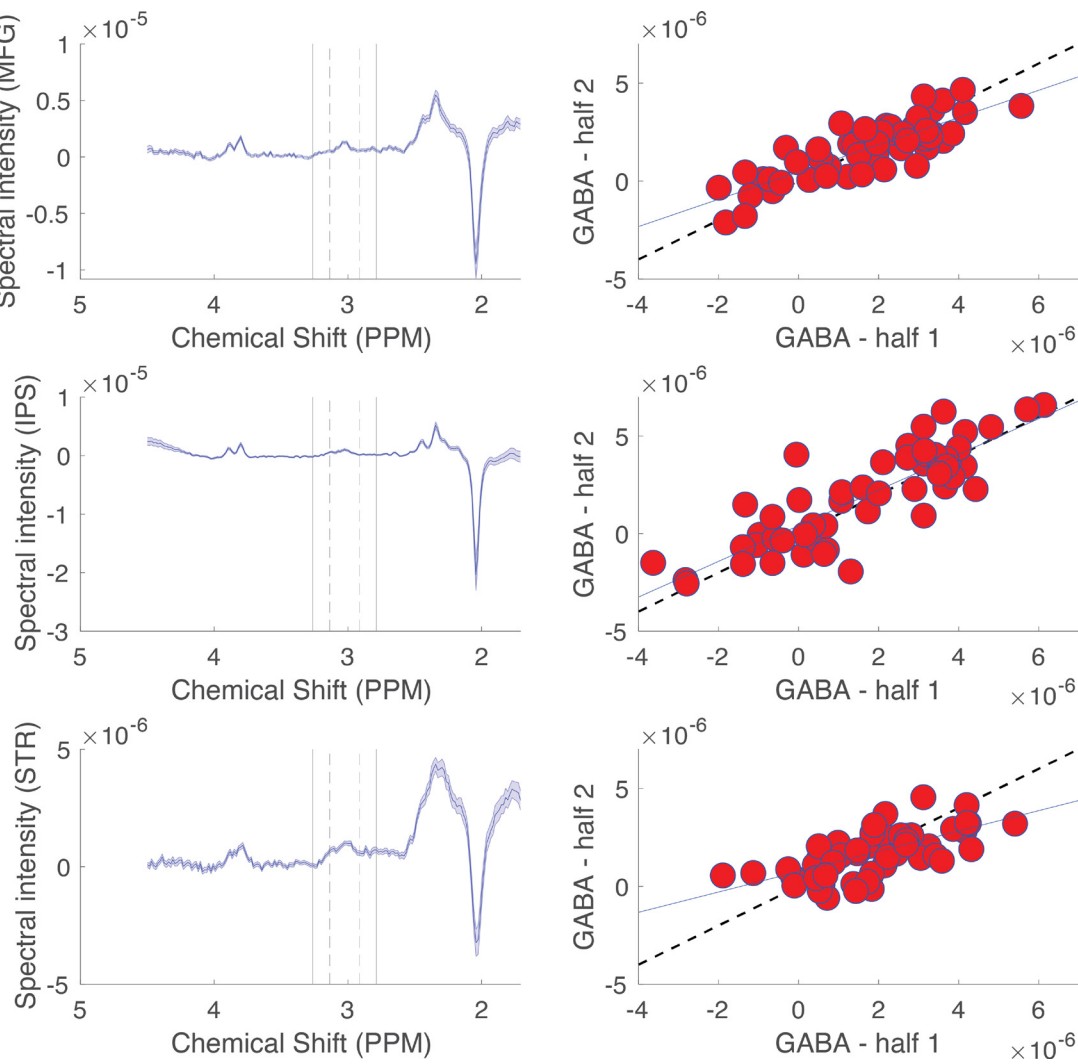

**Appendix 1—figure 6.** Description and validation of GABA quantification methods. The GABA peak in megaPRESS difference spectra was quantified by integrating chemical shifts ranging from 2.9 to 3.1 ppm and subtracting out the integrated signal in a surrounding reference window (2.8–2.9, 3.1–3.2 ppm). The reliability of this quantification method in our dataset was tested by splitting megaPRESS data into two halves, the first corresponding to the 'odd' difference spectra and the second corresponding to the 'even' difference spectra. Split half correlations between peak integrals for the two halves served to measure the reliability of our analysis method. Left: averaged difference spectra for the three brain regions: middle frontal gyrus (MFG, top), intraparietal sulcus (IPS, middle), and striatum (STR, bottom). Dotted vertical lines mark the edges of the integration window, whereas solid vertical lines mark the edges of the reference window. Right: split half correlations for each brain region (top = MFG, middle = IPS, bottom = STR) were relatively high ($R$ = 0.8, 0.8, and 0.68, respectively, for the three regions). GABA quantification on even trials (ordinate) is plotted against GABA quantification on odd trials (abscissa) for each participant (red points). Dotted line reflects the unity line, and solid line reflects a least-squares linear fit.

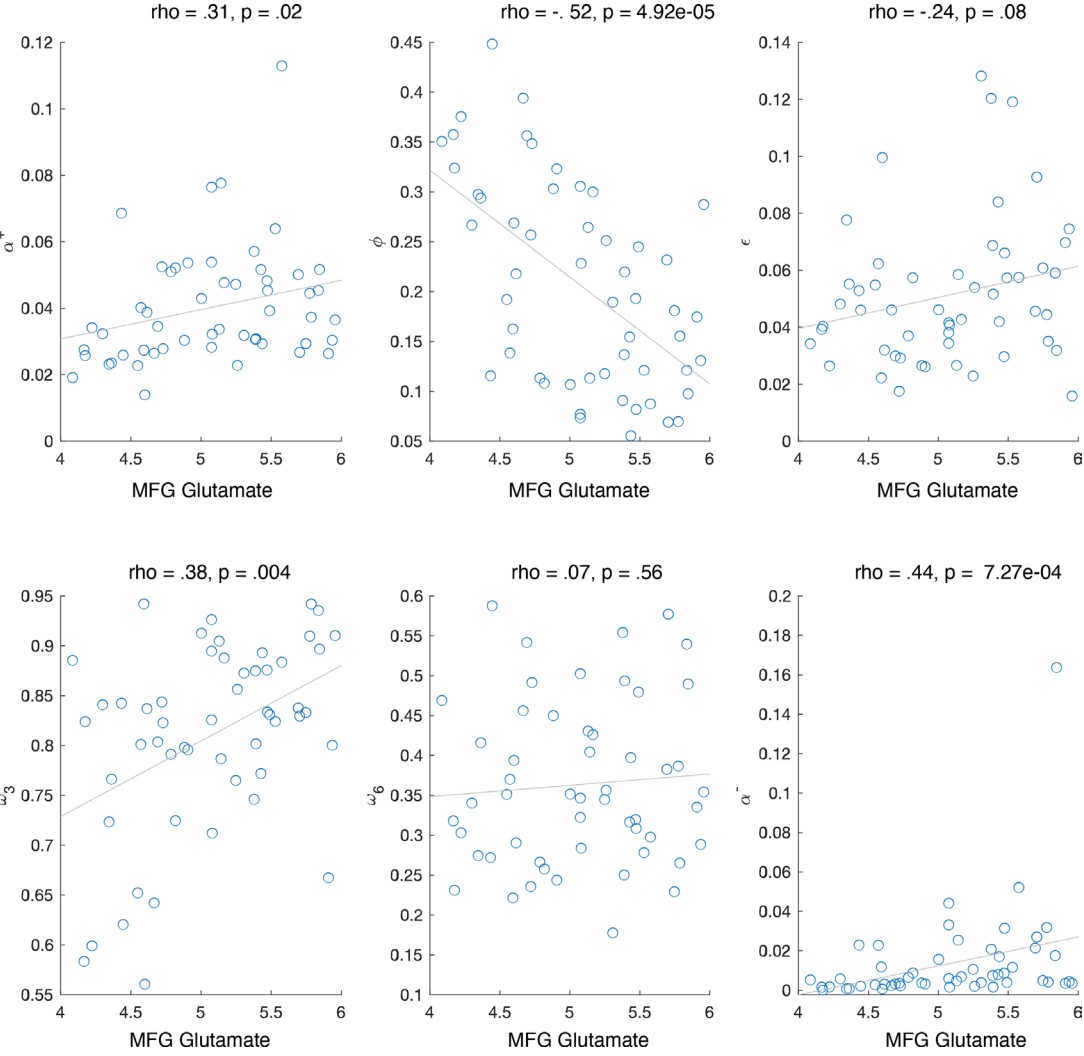

**Appendix 1—figure 7.** Correlation between middle frontal gyrus (MFG) glutamate and model parameters. The working memory (WM) parameters $\phi$ and $\omega_3$ correlated with MFG glutamate; reinforcement learning (RL) parameter $\alpha$ also correlated with MFG glutamate, but not when controlling for $\phi$. Negative learning rate $\alpha^-$ also correlated with MFG glutamate. Note that negative learning rate is included in updating of both RL and WM, and thus cannot be used to make claims about the specificity of WM or RL relationship with MFG glutamate.

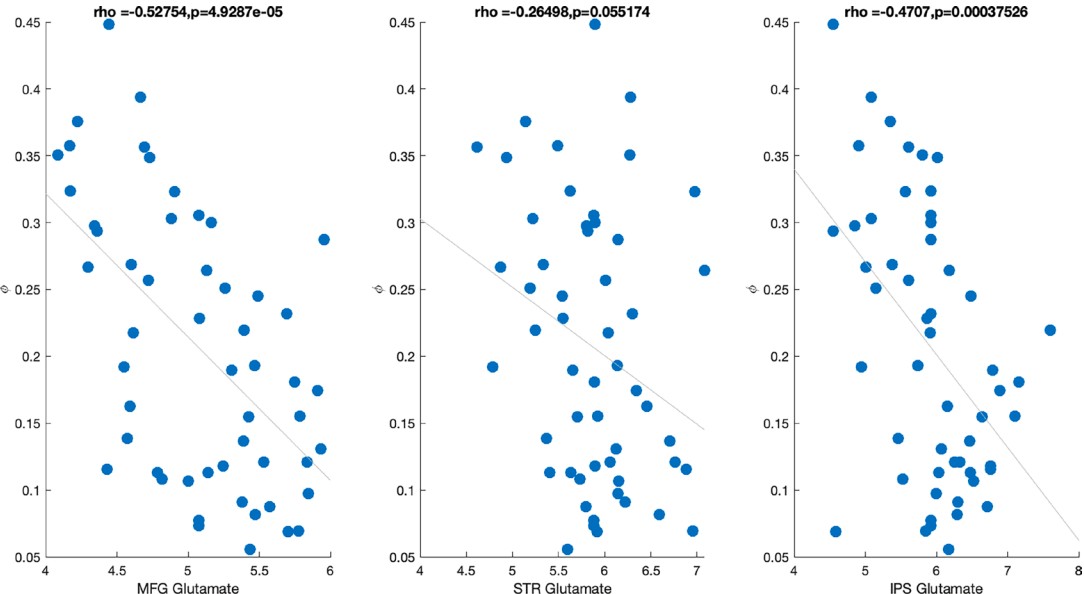

**Appendix 1—figure 8.** Correlation between working memory (WM) decay parameter ($\phi$) and glutamate sampled from three different brain regions (middle frontal gyrus [MFG], striatum [STR], and intraparietal sulcus [IPS]). Both MFG and IPS glutamate were significantly correlated with WM decay; when all three measures are entered as predictors of WM decay in the same model only MFG glutamate remains a significant predictor (see text).

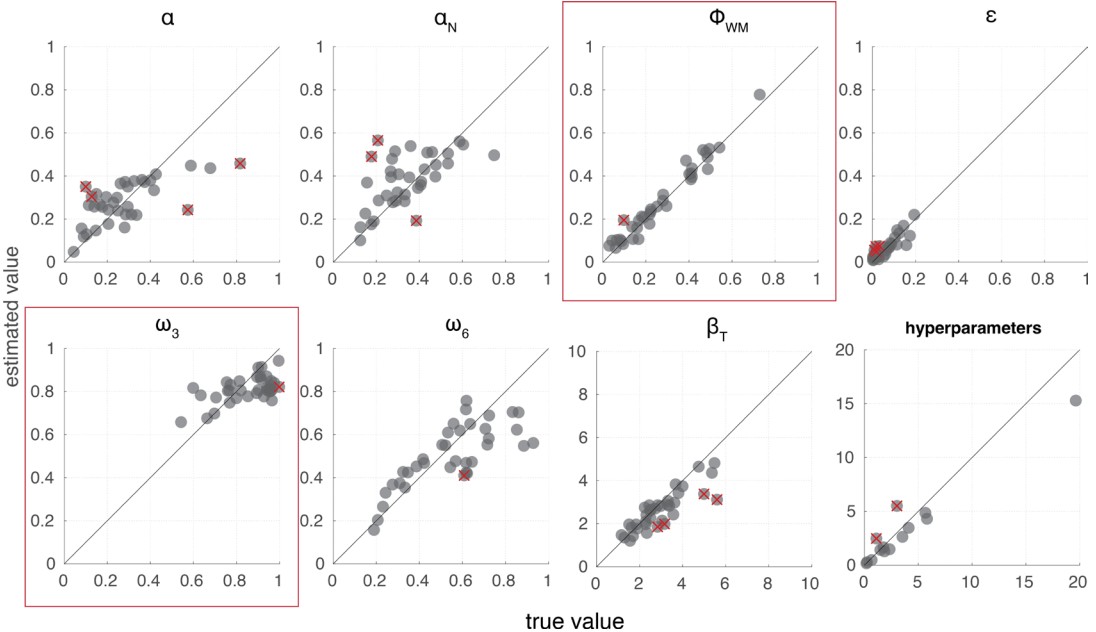

**Appendix 1—figure 9.** Reinforcement learning–working memory (RL-WM) model parameter recovery. We ran a recovery study to demonstrate the ability of the model to generate parameter estimates that correspond usefully to known, true parameter values. Data were simulated according to the RL-WM model specification using the same number of blocks, trials, etc., as in our experiment. The number of simulated participants matched the minimum number of participants in either of the two age groups (i.e. 36 simulated participants). Overall, 92.5% of parameters' credible intervals included the true value; both individual-level parameters and group-level hyperparameters were able to be recovered successfully.

## Mediation analysis

Since we observed age-related difference in glutamate and WM decay, and given that our dataset is not longitudinal and we cannot make claims with regards to causality direction in observed effects,

we have considered the questions of whether (1) effect of age on WM is mediated by glutamate, and (2) whether the effect of WM on task performance is mediated by glutamate.

To address the first question, we ran the mediation analysis looking at whether the effect of age on WM decay is mediated by glutamate. We found that the average direct effect (ADE), which is a direct effect of age on WM decay, accounting for the indirect effect of glutamate was significant ($ADE = 0.063, p = < 2e - 16$; ADE is equal to the $\beta_1$ coefficient from the following model: decay $= \beta_0 + \beta_1$ Age $+ \beta_2$ Glutamate). The average causal mediation effect (ACME) was not significant ($ACME = 0.009, p = 0.28$), with proportion mediated estimated at 0.13. ACME is equal to the product of coefficients $\beta_1$ and $\beta_2$ from the following two models: glutamate $= \beta_0 + \beta_1$ Age and decay $= \beta_0 + \beta_1$ Age $+ \beta_2$ Glutamate; both coefficients are negative because age is inversely correlated with glutamate, and decay is inversely correlated with glutamate, thus providing a positive ACME. The typical conclusion from these results would suggest that age can lead to a global deterioration of the brain, that is in part reflected in the age-based glutamate difference, and that subsequently impacts WM mechanisms. However, this interpretation is weakened by the fact that mediation analyses, like other tests that involve statistically controlling for variables, are critically sensitive to the measurement noise on those variables (*Westfall and Yarkoni, 2016*). In our case, we have accurate measures of age, but are relying on indirect measures of glutamate that involve considerable noise. This concern is further amplified by general power considerations for mediation analyses (*Fritz and Mackinnon, 2007*). Thus, while we were unable to reject the null hypothesis (that glutamate does not mediate the effects of age on working memory), we are also not confident in our ability to rule out a causal mediation effect.

