## [Editor Report]

This important study combines behavior, computational modelling and magnetic resonance spectroscopy in a cross-sectional design to address the question of whether age-related differences in learning are driven by changes in working memory decay or deficiencies in the reinforcement learning (RL) system. The general approach is convincing, the data novel, and the analysis carefully executed. Future work requires a longitudinal design to separate aging from cohort effects and may address the generality of the effects to other RL/Working memory tasks.

---

## [Decision Letter]

**Decision letter after peer review:**

Thank you for submitting your article "Age-related decline in prefrontal glutamate predicts failure to efficiently deploy working memory in the service of learning" for consideration by *eLife*. Your article has been reviewed by 3 peer reviewers, including Claire M Gillan as the Reviewing Editor and Reviewer #1, and the evaluation has been overseen by Jonathan Roiser as the Senior Editor.

Essential revisions:

We all enjoyed reading this paper, but all agreed that there were areas where the main claims perhaps went beyond the data. We hope that you find these recommendations helpful in strengthening the clarity of the paper and helping the reader take away what they need to.

1) The causal language (including in the title) should be tempered. Phrases such as 'is associated with' are preferred over 'predicts' given the nature of the data. Similarly, the reference to a 'decline' implies longitudinal data, where it is in fact a cross-sectional design.

2) The reviewers noted that no RL deficits were observed in this paper, so we did not feel it was justified to make broad claims about WM deficits 'explaining them' more generally. This bigger idea would need to be supported with data from a suite of RL tasks and would necessitate the basic observation of an RL deficit in older adults. We do not mean to say that additional data is required for publication, but absent it, the main claims need to be substantially toned down and qualified.

3) A mediation analysis is desirable to strengthen the contention that glutamate changes are responsible for WM deficits in older adults.

4) All reviewers queried the strength of the evidence for a working memory deficit – there was a change in decay but not deployment – can the authors support the interpretation of this as a deficit in efficient deployment of WM?

*Reviewer #1 (Recommendations for the authors):*

With respect to the computational modeling results, it would be helpful to explain to the reader (and me) a bit more about how the older adults can have faster decay of memory but it does not influence the trade-off with RL (ω parameter). Is that interpreted as – given their poorer memory / faster decay, they weight 'what they have' just as well as others. And if the model did not feature a decay (or fixed it), one would presume a difference in the weight would then be apparent. Is this the correct way to think about that result?

It appears to be a limitation that older adults were from the community while younger adults were from the Brown University participant pool. I understand the RBANs was taken, but absent major cognitive decline this is a coarse instrument (i.e. has ceiling effects in healthy samples). It does not appear there were any controls for level of education or IQ which might reasonably systemically vary across these samples. If this is the case, this may present an additional confound.

The section where the best fitting model is used to generate brain based predictions of performance was a bit hard to follow. It appears though that the effects of WM params are not just from glutamate but the other non-significant neural measures too. Why was this done? It seems the value of the two step analysis process is to identify the key contributor to performance (glutamate) and then focus on it? "WM parameters φ and ω capture the performance predictions based on glutamate (and remaining non-significant neural measures)"

The final section looking at how age factors into the relationship between WM and glutamate is interesting, but I wonder if it would be strengthened with a formal mediation model.

*Reviewer #2 (Recommendations for the authors):*

Included here are some analytical suggestions for addressing some of the points I raised in the public review.

1. The first point should be easy to address by conducting the same analysis using learning rates as the predicted variable [point 1].

2. To argue for specificity to *prefrontal* glutamate, it would be important to show Figure 3D including the STR measurement. The authors included that information in the text, but it's easy to miss especially because brain measurements were initially averaged [point 1].

3. For 3D, why not run a mixed effects model with Age Group, Glutamate, and Region regressed on WM decay? That seems like a more direct test of the question of whether PFC glutamate predicts WM decay, and whether this relationship interacts with age [point 5].

4. Provided the Age x Glutamate interaction effect above is significant (that is, glutamate predicts WM decay only in the OA group), it would strengthen the conclusion of the study to see a formal mediation analysis within the OA group, asking whether the WM effect on task performance is indeed modulated by glutamate [point 5].

*Reviewer #3 (Recommendations for the authors):*

1. The paper could benefit from a better description of the rationale for how RL versus WM contributions to learning are disentagled. The statements about the interpretation of set size difference Mentioned in the public review seem surprising in light of their little role in the papers conclusions.

1a. The authors do not statistically test the age-by-condition difference, which would be important to know. Judging from the plots, age differences in learning performance by condition seem biggest in the set size 6 condition, suggesting that in fact gradual RL computations are impaired quite markedly.

2. The rationale for emphasising the difference between learning and test performance was unclear too me. Doesn't the 20 min delay between learning and test suggest that what we see there is related to LTM consolidation, rather than WM capacity? In addition, the set size 3 vs 6 condition difference by age was marginal.

3. The statical approaches could be better explained and motivated. It was unsure why the analysis of model parameters didn't include p values or test statistics. Likewise, the motivation behind the analysis that investigates the factors that relate to the brain/MRS predicted task performance was unclear to me.

4. The task could be described in a bit more detail, in particular the test phase. The fact that this is only binary feedback rather than a

5. Does the fitting of β on the group level disadvantage one of the age groups? Or was this done separately for each group?

6. Which regressors were part of which model in the MRS analysis remained unclear. I was wondering in particular about issues of collinearity, which can impact the ability to interpret the β coefficients (which the paper relies on, rather then the model comparisons)

7. In the section investigating the relation between age, WM decay and glutamate, it be useful to see how strong the relation between glutamate and WM is after age has been factored out.

---

## [Author Response]

Essential revisions:We all enjoyed reading this paper, but all agreed that there were areas where the main claims perhaps went beyond the data. We hope that you find these recommendations helpful in strengthening the clarity of the paper and helping the reader take away what they need to.1) The causal language (including in the title) should be tempered. Phrases such as 'is associated with' are preferred over 'predicts' given the nature of the data. Similarly, the reference to a 'decline' implies longitudinal data, where it is in fact a cross-sectional design.

Thank you for your suggestion. We agree that since we cannot establish causality in our data, the language throughout the manuscript should reflect the associative link between the behavioral/computational/neural mechanisms (and similarly for the cross-sectional nature of the project). We have adjusted the title and the language throughout the manuscript accordingly.

2) The reviewers noted that no RL deficits were observed in this paper, so we did not feel it was justified to make broad claims about WM deficits 'explaining them' more generally. This bigger idea would need to be supported with data from a suite of RL tasks and would necessitate the basic observation of an RL deficit in older adults. We do not mean to say that additional data is required for publication, but absent it, the main claims need to be substantially toned down and qualified.

Thank you for pointing this out. We believe that this is largely a semantics issue, and we fully agree that we should make our wording more clear. The confusion pertaining to our claims of WM explaining performance deficits may arise because the term “RL” (confusingly) refers to three different concepts: (1) the RL task/behavior, (2) the RL process component of the model, and (3) the neural RL mechanism. Our task is what the literature calls an RL task (participants learn correct stimulus-action associations from rewards), with an added manipulation targeted at revealing WM. We observe a deficit in behavior in older adults, the “RL” deficit, where RL refers to “behavior in an RL task”. Our model separates two processes, RL and WM, that simultaneously contribute to the task behavior. This allows us to determine if the observed patterns of behavior in the RL task should indeed be attributed to an RL process, or to an alternative process (WM; Collins and Frank, 2012). We observe an impairment in the WM process primarily, such that we can conclude that WM memory deficits (not RL process deficits) underlie the RL behavior deficit. We believe that there is a distinction between RL behavior (learning from rewards), and what RL means in the context of our model. Specifically, the RL algorithm means to capture incremental learning that is not capacity limited, analogous the RL thought to be implemented by dopaminergic processes in the brain. Therefore, while on the surface level the deficit in behavior may appear as RL-driven (e.g. slower learning from rewards), there may *not* be an impairment in the underlying RL algorithm and its neural basis, as indicated by our model We have adjusted the language in the paper to make these points more clear, and to reflect the suggestions of the reviewers.

We agree that we cannot make broader claims regarding all types of RL behavior, as different RL tasks may recruit different underlying processes, including RL and WM. Indeed, our recent work (Eckstein et al., 2022) shows limited generalizability across RL tasks. We now include this in the discussion.

3) A mediation analysis is desirable to strengthen the contention that glutamate changes are responsible for WM deficits in older adults.

We appreciate your suggestion. We have understood from 2 reviewers that two different kinds of mediation approaches would be of interest. First would apply to running a mediation analysis including WM decay, age, and glutamate with the goal of testing how much of the effect of age on WM performance is mediated by glutamate (suggested by Reviewer 1). Second, we would consider how much glutamate mediates WM effects on performance (suggested by Reviewer 2).

In response to the first question, we have run the mediation analysis looking at whether the effect of age on WM decay is mediated by glutamate. We found that the average direct effect (ADE), which is a direct effect of age on WM decay, accounting for the indirect effect of glutamate was significant (ADE = 0.063, p = <2e-16; ADE is equal to the β_1_ coefficient from the following model: decay = β_0_ + β_1_Age + β_2_Glutamate). The average causal mediation effect (ACME), was not significant (ACME = 0.009, p = 0.28), with proportion mediated estimated at.13. ACME is equal to the product of coefficients β_1a_ and β_2b_ from the following 2 models:

Glutamate = β_0a_ + β_1a_ Age and Decay = β_0b_ + β_1b_Age + β_2b_Glutamate; both coefficients are negative because age is inversely correlated with glutamate, and decay is inversely correlated with glutamate, thus providing a positive ACME. One possible conclusion from our failure to reject the null hypothesis is that age can lead to a global deterioration of the brain, that is in part reflected in the age-related glutamate difference, and that subsequently impacts WM mechanisms. However, this interpretation is muddied by the fact that mediation analyses, like other tests that involve statistically controlling for variables, are critically sensitive to the measurement noise on those variables (Westfall and Yarkoni). In our case, we have accurate measures of age, but are relying on indirect measures of glutamate that involve considerable noise. This concern is further amplified by general power considerations for mediation analyses (Fritz and MacKinnon, 2010). Thus, while we were unable to reject the null hypothesis (that glutamate does not mediate the effects of age on working memory) we are also not confident in our ability to rule out a causal mediation effect. With these things in mind, we have now included these results in our paper, along with the interpretation above and our concerns with that interpretation.We have also gone back to carefully change language in places where our previous manuscript implied a mediation effect. It is noteworthy that an alternative version of this mediation model where age is used as a mediator would violate the core assumption of mediation analysis – the sequential nature of the relationship between predictor, mediator and the outcome (X -> M -> Y; Cole and Maxwell, 2003), because age precedes both glutamate and WM decay (M -> X and M -> Y) and thus would not be appropriate. We include this information because we were not exactly sure what mediation analysis was requested and it is possible that this is the model that the reviewer was looking for.

In response to the second suggestion, we have run a second mediation model that tests whether/how much glutamate mediates the effect of WM on performance. We have entered WM as a predictor, glutamate as mediator in a model predicting average task performance. The results show that ADE (effect of WM on performance while controlling for effect of glutamate) was significant (ADE = -.014, p <2e-16). ACME (product of β_1_ and β_2_ coefficients from following 2 models: Glutamate = β_0_ + β_1_WMdecay and performance = β_0_ + β_1_WMdecay + β_2_Glutamate) was also significant (ACME = -.01, p = .008), with proportion mediated estimated at.19. This suggests that glutamate mediates the effect of WM memory decay on performance.

However, we would like to note an issue with this analysis: WM decay is estimated from task performance – thus, WM decay and task performance share signal as well as noise. We repeated the mediation analysis, but substituted WM decay with participants’ immediate memory score from RBANS battery, in order to use a RL-WM task and model independent index of WM. However, this analysis approach was limited by the fact that the working memory measures extracted from the RBANS were not tightly linked to task behavior. In this model, ADE and ACME were both not significant (ADE = 2.09e-02, p = .06; ACME = -5.93e-05, p=.96). We also ran a mediation analysis with total RBANS score instead of immediate memory, as previous work showed correlation between WM and total RBANs score (Gold et al.,1999). But again, neither ADE or ACME from this mediation were significantly different from zero (ADE = .005, p = .70; ACME = .01, p=.14). Note that immediate memory and total score from RBANS likely capture WM aspects such as capacity more than WM decay (which highlights the utility of computational modeling for specific effects), which could explain the absence of direct effect of these metrics on performance, even when mediator is not included.

Again, it is worth highlighting the power limitation of our data; mediation analysis requires a considerable sample size to achieve adequate power (Fritz and MacKinnon, 2010), with reliable mediation results requiring well over hundred participants. Therefore, the conclusions we can draw from our mediation results are limited. Nonetheless, we have now included the discussion of the age-related mediation approach in Appendix 1, results and discussion, because we have strong concerns with the two versions of the mediation that test whether glutamate mediates WM effect on performance(in version 1, because performance was used to attain WM measures, in version 2, because there was no direct effect of WM measures on performance to mediate).

4) All reviewers queried the strength of the evidence for a working memory deficit – there was a change in decay but not deployment – can the authors support the interpretation of this as a deficit in efficient deployment of WM?

Thank you for pointing this out. We think of working memory as a multifaceted cognitive function, that is defined by multiple components – resource or capacity (e.g. how much information/ how many items can be stored in WM), decay of information held in WM, deployment of WM in service of the task, etc. As such, WM impairment need not be global some components of WM may be impaired while others remain intact. In our case, the deployment of WM was preserved, while decay was larger in older adults. We now further elaborate on this on lines 351-353 in the Discussion section. We also acknowledge that the title of our manuscript may have contributed to this confusion (implying impairment in deployment WM, while results reveal intact deployment and impaired decay), and we propose changing the title of the manuscript to “Age-related differences in prefrontal glutamate are associated with increased working memory decay that gives appearance of learning deficits”.

Reviewer #1 (Recommendations for the authors):With respect to the computational modeling results, it would be helpful to explain to the reader (and me) a bit more about how the older adults can have faster decay of memory but it does not influence the trade-off with RL (ω parameter). Is that interpreted as – given their poorer memory / faster decay, they weight 'what they have' just as well as others. And if the model did not feature a decay (or fixed it), one would presume a difference in the weight would then be apparent. Is this the correct way to think about that result?

Thank you for the comment. As mentioned in the response to essential revisions, our model formalizes WM as a multifaceted process defined by multiple parameters (here decay and WM-reliance parameter). It is plausible for some aspects of WM to be impaired (in this case decay), white others remain intact. Your interpretation is correct – if we excluded WM decay from the model (or fixed it), then indeed the group difference might shift to other WM parameters (or may be absorbed by RL parameters). However, we have confirmed that WM decay and WM weight are separable using model parameter recovery (see figure below; we have now included it in Appendix 1).

Furthermore, we showed that by fixing WM decay in the old adults group (by setting it to the mean decay of the young adults group) and simulating synthetic datasets we can fix the observed performance deficit (see Appendix 1 – Figure 5). Put another way, the impaired performance of older adults in the task was best fit by considering them to use a less reliable working memory system, but relying on it to a similar degree as the younger adults. We have now tried to make this point clear in our results and discussion.

It appears to be a limitation that older adults were from the community while younger adults were from the Brown University participant pool. I understand the RBANs was taken, but absent major cognitive decline this is a coarse instrument (i.e. has ceiling effects in healthy samples). It does not appear there were any controls for level of education or IQ which might reasonably systemically vary across these samples. If this is the case, this may present an additional confound.

Thank you for the comment. To our knowledge, RBANS is extensively used to evaluate individuals with normal cognition (Cooley et al., 2015; Duff and Ramezani, 2015; Thaler et al., 2015), and is not known to have ceiling effect issues, which is more common in measures such as MMSE or MoCA on which cognitively healthy individuals tend to perform at near ceiling. The RBANS was designed to increase sensitivity to small cognitive differences. The RBANS total score, which we compared across groups, is highly correlated with IQ (R=0.77, Gold et al., 1999). Age normalized RBANS data from our participants show no signs of ceiling level performance, indeed the distributions, as shown in Author response image 1, indicate a relatively normal distribution and no clear differences across groups. Thus, while we do not deny that cohort differences exist across our groups, we do not believe that our cohorts reflect different population samples in terms of overall cognitive ability.

**Author response image 1. sa2fig1:** 

The section where the best fitting model is used to generate brain based predictions of performance was a bit hard to follow. It appears though that the effects of WM params are not just from glutamate but the other non-significant neural measures too. Why was this done? It seems the value of the two step analysis process is to identify the key contributor to performance (glutamate) and then focus on it? "WM parameters φ and ω capture the performance predictions based on glutamate (and remaining non-significant neural measures)"

We apologize for the confusion. We focused on identifying *the best model* that included neural measures to predict behavioral performance; the model with glutamate and additional factors generated better out of sample predictions than the model that only contained glutamate – thus we generated the performance predictions based on this full model. Once we had identified the best predictive model, we then examined which predictors were significantly different from zero within the model, and found that this was only the case for glutamate. Put simply, we first identified the best model, then looked to see what factors in that model were doing the heavy lifting, and identified glutamate. We then focused on glutamate alone to explore the anatomical specificity of the relationship between glutamate and model parameters. For completeness, we now also have run a model predicting WM decay with the set of all predictors from the best model, and found that none of the predictors related to WM when controlling for glutamate (GABA t = -.60, p = .55; NAA t = – 0.94, p = .34; CMF cortical thickness t = -0.90, p = 0.37; SF cortical thickness t = -.78, p = .43). We have further clarified these points in the analysis and Results sections.

The final section looking at how age factors into the relationship between WM and glutamate is interesting, but I wonder if it would be strengthened with a formal mediation model.

We have now conducted two classes of mediation analyses as described in our response to editorial point #3. While the results of these analyses are somewhat difficult to interpret, we now refer to them in the results and include them in Appendix 1.

Reviewer #2 (Recommendations for the authors):1. The first point should be easy to address by conducting the same analysis using learning rates as the predicted variable [point 1].

Thank you. We have included the figure in the response above, and also in the Appendix 1.

2. To argue for specificity to *prefrontal* glutamate, it would be important to show Figure 3D including the STR measurement. The authors included that information in the text, but it's easy to miss especially because brain measurements were initially averaged [point 1].

Thank you for the suggestion. We have included the figure showing correlation between STR glutamate measurement and decay in the Appendix 1 (see Appendix 1-figure 8). Note that while the correlation for IPS glutamate is significant, and correlation for STR glutamate is on the margin of being significant, when we control for MFG glutamate only MFG glutamate remains a significant predictor (as described in the text).

3. For 3D, why not run a mixed effects model with Age Group, Glutamate, and Region regressed on WM decay? That seems like a more direct test of the question of whether PFC glutamate predicts WM decay, and whether this relationship interacts with age [point 5].4. Provided the Age x Glutamate interaction effect above is significant (that is, glutamate predicts WM decay only in the OA group), it would strengthen the conclusion of the study to see a formal mediation analysis within the OA group, asking whether the WM effect on task performance is indeed modulated by glutamate [point 5].

Response to points 3 and 4: We (1) ran an interaction model predicting WM decay using age-by-MFG glutamate interaction, and (2) correlation between MFG glutamate and WM decay in the two age groups separately. The interaction was not significant (β = -.01,t = .69, p=.48). We found that glutamate did not correlate with WM decay in young adults (r=.11, p = .54), but there was a trending correlation between glutamate and WM decay in older adults (r = -.36, p = .08). While our inability to detect a significant interaction might be in part driven by the limited sample size, we consider that inverse correlation between glutamate and WM decay across the sample, along with trending negative correlation between glutamate and decay within old adults suggest that prefrontal glutamate levels are lower in older adults group relative to young adults group, and thus contribute to age-related differences in working memory that may account for observed learning impairments.

Reviewer #3 (Recommendations for the authors):1. The paper could benefit from a better description of the rationale for how RL versus WM contributions to learning are disentagled. The statements about the interpretation of set size difference Mentioned in the public review seem surprising in light of their little role in the papers conclusions.

Thank you for the suggestion. We have now included more detailed comments on how the model helps us disentangle RL and WM in the Computational model part of the methods section. With regards to your comment about the set size difference and how it relates to our results that indicate the difference in WM, as stated in the earlier response: higher set size assumes increased delay between successive presentations of the same stimuli. Longer delays, in turn, make WM forgetting more pronounced. Therefore, the modeling results are not necessarily inconsistent with the greater age group difference in set size 6 condition compared to set size 3 condition. This simply underscores the importance of modeling, as it allows us to identify the mechanisms that drive behavioral patterns with greater specificity. Furthermore, we simulated data from parameters estimated for older adults, with one key difference – WM decay was set to the group mean value of the younger adult group. The simulated performance shows that fixing WM decay to a smaller value in older adults rescues performance – primarily in the set size 6 condition (See Results section and Appendix 1 – Figure 5).

1a. The authors do not statistically test the age-by-condition difference, which would be important to know. Judging from the plots, age differences in learning performance by condition seem biggest in the set size 6 condition, suggesting that in fact gradual RL computations are impaired quite markedly.

We have tested for group difference in set size, (2-sample t-test set size 3: t(76) = 4.71, p = 1.05e-05 ; 2-sample t-test set size 6: t(76) = 7.03, p = 7.61e-10 – see the Results section). Furthermore, we tested the age effect on the difference between set size conditions, which also showed that the difference between set size 6 and set size 3 performance is greater in old adults (t(76) = 4.12, p = 9.55e-05), which also supports the reviewer’s interpretation that age difference in learning performance by condition are bigger in set size 6. This, however, does not necessarily mean that RL computations are more impaired in the older adults group, even though on the surface it may appear to be the case (hence the need for computational modeling). As stated in the response to the previous point, fixing WM decay in old adults by setting it to the group mean value of younger adults (thus effectively reducing WM decay) for each older adult participant rescues their performance in set size 6 condition (see results lines 235-240 and also Appendix 1 – Figure 5). Thus, while aging impairments were numerically larger in the large set size condition where RL typically has a larger role, they were best explained computationally by a working memory impairment, and “fixing” this impairment was sufficient to rescue large set size task performance in our model.

2. The rationale for emphasising the difference between learning and test performance was unclear too me. Doesn't the 20 min delay between learning and test suggest that what we see there is related to LTM consolidation, rather than WM capacity? In addition, the set size 3 vs 6 condition difference by age was marginal.

Thank you for the question. First, in regards to the second point, the reviewer is correct that WM cannot support performance in the test phase, where no new information is provided (no feedback), and past information is too far in the past (delayed by the irrelevant task participants perform between learning and testing). We hypothesize that the test phase reflects stimulus-action associations encoded in a long-term fashion by the brain’s RL system; however, it is possible that other forms of long-term storage of information (such as episodic memory) also play a role (Bornstein and Norman,2017). We address this in the task section of the methods.

In regards to the rationale for the test phase: As explained above, compared to the learning phase where WM highly contributes to performance, the test phase reflects better isolated slow, long-term learning processes. Investigating test phase performance in comparison to learning phase performance allows us to see how learning within the RL process is impacted by use of WM during learning. We replicate the previous finding that using WM during learning speeds up learning at the cost of decreasing long term retention in RL, showing an interaction between the two learning processes. We agree that this interaction effect is not significantly different between age groups, and consequently we do not conclude that the WM impacts learning within the RL processes differently. We have now clarified these points in the methods and referenced previous work that has gone into greater detail with respect to set size and test phase manipulations.

3. The statical approaches could be better explained and motivated. It was unsure why the analysis of model parameters didn't include p values or test statistics. Likewise, the motivation behind the analysis that investigates the factors that relate to the brain/MRS predicted task performance was unclear to me.

Thank you for your comment. We have not reported p-values or other frequentist statistics in model-based analysis, because we have relied on a Bayesian approach to estimate the differences in parameters between age groups. In Bayesian statistics, using p-values as measures of evidence is not recommended (Kruschke, 2014); therefore, we used credible intervals instead. As is common in the Bayesian cognitive modeling literature, we interpret credible intervals that exclude 0 as support for a conclusion that the estimated difference in parameter values is not zero.

With regards to our data-driven approach in linking MRS measures to model parameters and behavior, we outline each step and the motivation behind each step below. The main motivation for implementing this approach was to reduce the dimensionality of our highly-correlated data, as well as the number of statistical tests we’d otherwise need to perform for different measures/model parameters. A more detailed description is now included in the methods section, as well as the relevant Results section.

Step 1: Average measures across regions, for all measures that were sampled from multiple regions (Motivation: reduce the number of features in a meaningful way – e.g. glutamate is an average of only glutamate measures from all the areas glutamate measures were sampled from).

Step 2: Implement cross-validated model selection (Motivation: cross-validation is a commonly used method in machine learning, used to select a model with features that maximizes accuracy of out-of-sample predictions, and model selection allows us to narrow brain measures to only those that can contribute usefully to behavioral predictions).

Step 3: Generate performance predictions based on the set of features from the best model from Step 2, and then regress model parameters onto this prediction (Motivation: the prediction represents the projection of brain measures onto a single learning behavior and the alternative approach of correlating *each* brain measure with performance would increase the number of tests, and thus our expected rate of false positives)

Step 4: Identify which parameters within “best” model of behavior take values that differ significantly from zero (Motivation: identify most promising parameters to avoid running multiple tests for all parameters, even those explaining a small amount of variance; this also allowed us to estimate the effect of each parameter in predicting the brain-generated performance prediction, while controlling for other parameters).

Step 5: Identify the significant brain measure predictor from, and explore how anatomically specific measures relate to model parameters from Step 4 (Motivation: once we have narrowed down the explanatory brain measures and model parameters, we can explore the anatomical and computational specificity of the observed relationships).

4. The task could be described in a bit more detail, in particular the test phase. The fact that this is only binary feedback rather than a

It seems like the reviewer’s comment was cut here. In case the reviewer was asking about the difference between binary feedback and a random number of points as feedback we have now clarified this in the paper. In response to the first point, we have reviewed and extended the test phase task description.

5. Does the fitting of β on the group level disadvantage one of the age groups? Or was this done separately for each group?

The latter is correct. Our model specification included two separate learning phase β parameters, one for the young age group and one for the older age group. (Estimates are shown in the second to last panel of Figure 2C.)

6. Which regressors were part of which model in the MRS analysis remained unclear. I was wondering in particular about issues of collinearity, which can impact the ability to interpret the β coefficients (which the paper relies on, rather then the model comparisons)

Thank you for your comment. We have now modified the Results section to list regressors for each of the models we ran in MRS analysis in a more clear way. To address the collinearity concern, the cross-validation approach in which we first averaged each measure across the regions they were sampled from was implemented with an intent of reducing the number of collinear measures. To demonstrate this, in Author response table 1 we report the variance inflation factors (VIF, which quantifies how much the variance of estimated coefficients is inflated due to collinearity , with large VIFs indicating increased inflation/collinearity) for the set of predictors prior to passing them through the cross validation pipeline (each has 3 areas, all were included for the purpose of computing variance factors) :

**Author response table 1. sa2table1:** 

	IPS	MFG	STR
GABA	1.82	1.57	1.43
Glutamate	4.60	3.24	1.50
NAA	4.53	2.85	2.07
Gray matter	12.70	20.78	28.01
White matter	10.27	17.11	28.98
	CMF	SF	RMF
Cortical thickness	3.98	4.01	5.44

As we can see, the VIFs for predictors are very high. Thus, if we had simply put all of our biological variables into a single behavioral regression, our power would have suffered massively due to collinearity in our predictors. This was a primary motivation for our analysis pipeline, as detailed above, which first combined measures across regions and then used model selection to reduce the number of predictors considered. In Author response table 2 we report VIFs for the best set of predictors, identified following averaging averaging and cross validation:

**Author response table 2. sa2table2:** 

GABA	1.16
Glutamate	2.06
NAA	1.76
CMF cortical thickness	2.71
SFcortical thickness	2.37

If we take the assumption that VIFs larger than 3 indicate cause for concern due to collinearity, we can see that pruning the set of predictors using the steps described in response to point 3 we identified the set of predictors with VIFs which do not indicate high collinearity. We now include the VIFs of the best set of predictors in the Results section.

7. In the section investigating the relation between age, WM decay and glutamate, it be useful to see how strong the relation between glutamate and WM is after age has been factored out.

We have run the model that includes an interaction (WM decay ~ 1 + MFG glutamate * Age), and did not find a significant interaction (interaction β = -.01, p = .48; likely in part due to sample size limitation). Furthermore, the main effects of MFG glutamate and age are also not significant when entered in the same model (MFG Glutamate β = -.02, p = .30, Age β = .14, p = . 21) because age and glutamate are correlated (r=-.59,p=2.93e-06). We have now included this in the Results section, and addressed it as a discussion point.

References

Collins, A. G., and Frank, M. J. (2012). How much of reinforcement learning is working memory, not reinforcement learning? A behavioral, computational, and neurogenetic analysis. European Journal of Neuroscience, 35(7), 1024-1035.

Eckstein, M. K., Master, S. L., Xia, L., Dahl, R. E., Wilbrecht, L., and Collins, A. G. (2022). The interpretation of computational model parameters depends on the context. *ELife*, 11, e75474.

Frank, M. J., and Kong, L. (2008). Learning to avoid in older age. Psychology and aging, 23(2), 392.

Hämmerer, D., Li, S.-C., Müller, V., and Lindenberger, U. (2011). Life span differences in electrophysiological correlates of monitoring gains and losses during probabilistic reinforcement learning. Journal of Cognitive Neuroscience, 23(3), 579–592.

Grogan, J., Isotalus, H., Howat, A., Irigoras Izagirre, N., Knight, L., and Coulthard, E. (2019). Levodopa does not affect expression of reinforcement learning in older adults. Scientific reports, 9(1), 1–10.

Samanez-Larkin, G. R., Worthy, D. A., Mata, R., McClure, S. M., and Knutson, B. (2014). Adult age differences in frontostriatal representation of prediction error but not reward outcome. Cognitive, Affective, and Behavioral Neuroscience, 14(2), 672–682.

Radulescu, A., Daniel, R., and Niv, Y. (2016). The effects of aging on the interaction between reinforcement learning and attention. Psychology and aging, 31(7), 747.

Randolph, C., Tierney, M. C., Mohr, E., and Chase, T. N. (1998). The repeatable battery for the assessment of neuropsychological status (rbans): Preliminary clinical validity. Journal of clinical and experimental neuropsychology, 20(3), 310–319.

Gold, J. M., Queern, C., Iannone, V. N., and Buchanan, R. W. (1999). Repeatable battery for the assessment of neuropsychological status as a screening test in schizophrenia, I: Sensitivity, reliability, and validity. American Journal of Psychiatry, 156(12), 1944-1950.

Cole, D. A., and Maxwell, S. E. (2003). Testing mediational models with longitudinal data:

questions and tips in the use of structural equation modeling. Journal of abnormal psychology, 112(4), 558

Bornstein, A. M., and Norman, K. A. (2017). Reinstated episodic context guides sampling-based decisions for reward. Nature neuroscience, 20(7), 997-1003.

Fritz, M. S., and MacKinnon, D. P. (2007). Required sample size to detect the mediated effect. Psychological science, 18(3), 233-239.

Westfall, J., and Yarkoni, T. (2016). Statistically controlling for confounding constructs is harder than you think. PloS one, 11(3), e0152719.

Kruschke, J. (2014). Doing Bayesian data analysis: A tutorial with R, JAGS, and Stan.

Bäckman, L., Nyberg, L., Lindenberger, U., Li, S. C., and Farde, L. (2006). The correlative triad among aging, dopamine, and cognition: current status and future prospects. Neuroscience and Biobehavioral Reviews, 30(6), 791-807.

O'Reilly, R. C., and Frank, M. J. (2006). Making working memory work: a computational model of learning in the prefrontal cortex and basal ganglia. Neural computation, 18(2), 283-328.

Cooley, S. A., Heaps, J. M., Bolzenius, J. D., Salminen, L. E., Baker, L. M., Scott, S. E., and Paul, R. H. (2015). Longitudinal change in performance on the Montreal Cognitive Assessment in older adults. The Clinical Neuropsychologist, 29(6), 824-835.

Duff, K., and Ramezani, A. (2015). Regression-based normative formulae for the repeatable battery for the assessment of neuropsychological status for older adults. Archives of Clinical Neuropsychology, 30(7), 600-604.

Thaler, N. S., Hill, B. D., Duff, K., Mold, J., and Scott, J. G. (2015). Repeatable Battery for the Assessment of Neuropsychological Status (RBANS) intraindividual variability in older adults: Associations with disease and mortality. Journal of clinical and experimental neuropsychology, 37(6), 622-629.

Li, S. C., Lindenberger, U., and Sikström, S. (2001). Aging cognition: from neuromodulation to representation. Trends in cognitive sciences, 5(11), 479-486.